# Improved SIFTER v2 algorithm for long-term GOME-2A satellite retrievals of fluorescence with a correction for instrument degradation

Erik van Schaik[1,a], Maurits L. Kooreman[2,a], Piet Stammes[2], L. Gijsbert Tilstra[2], Olaf N. E. Tuinder[2], Abram F. J. Sanders[2,4,5], Willem W. Verstraeten[1,2,3], Rüdiger Lang[5], Alessandra Cacciari[5], Joanna Joiner[6], Wouter Peters[1,7], and K. Folkert Boersma[1,2,a]

[a]These authors contributed equally to this work.

[1]Meteorology and Air Quality group, Wageningen University, 6700 AA Wageningen, The Netherlands
[2]Satellite Observations Department, Royal Netherlands Meteorological Institute, De Bilt, 3730 AE, The Netherlands
[3]Royal Meteorological Institute of Belgium (KMI), Ringlaan 3, B-1180, Ukkel, Belgium
[4]Institute of Environmental Physics, University of Bremen, Otto-Hahn-Allee 1, D-28359 Bremen, Germany
[5]EUMETSAT, EUMETSAT-Allee 1, D 64295 Darmstadt, Germany
[6]NASA Goddard Space Flight Center, Greenbelt, MD, United States
[7]University of Groningen, Centre for Isotope Research, 9747 AG Groningen, The Netherlands

*Correspondence to*: K. Folkert Boersma (folkert.boersma@wur.nl)

**Abstract.** Solar-induced fluorescence (SIF) data from satellites are increasingly used as a proxy for photosynthetic activity by vegetation, and as a constraint on gross primary production. Here we report on improvements in the algorithm to retrieve mid-morning (09:30 hrs local time) SIF estimates on the global scale from GOME-2 sensor on the Metop-A satellite (GOME-2A) for the period 2007-2019. Our new SIFTER (Sun-Induced Fluorescence of Terrestrial Ecosystems Retrieval) v2 algorithm improves over a previous version by using a narrower spectral window that avoids strong oxygen absorption and is less sensitive to water vapour absorption, by constructing stable reference spectra from a 6-year period (2007-2012) of atmospheric spectra over the Sahara, and by applying a latitude-dependent zero-level adjustment that accounts for biases in the data product. We generated stable, good-quality SIF retrievals between January 2007 and June 2013, when GOME-2A degradation in the near infrared was still limited. After the narrowing of the GOME-2A swath in July 2013, we characterized the throughput degradation of the level-1 data in order to derive reflectance corrections and apply these for the SIF retrievals between July 2013 and December 2018. SIFTER v2 data compares well with the independent NASA v2.8 data product. Especially in the evergreen tropics, SIFTER v2 no longer shows the underestimates against other satellite products that were seen in SIFTER v1. The new data product includes uncertainty estimates for individual observations, and is best used for mostly clear-sky scenes, and when spectral residuals remain below a certain spectral autocorrelation threshold. Our results support the use of SIFTER v2 data to be used as an independent constraint on photosynthetic activity on regional to global scales.

## 1 Introduction

Solar-induced fluorescence (SIF) by vegetation is directly related to light absorption by the chlorophyll complex during photosynthesis [Porcar-Castell et al., 2014; Mohammed et al., 2019]. Most of the solar energy that a plant receives is used for photosynthesis, but part is released as heat and between 1-2% is re-emitted as fluorescence at longer wavelengths [Baker and Oxborough, 2004]. The fluorescent light emitted by plants has a spectrally smooth signature with two peaks, one at 690 nm (red fluorescence) and one at 740 nm (far-red fluorescence) (e.g. Daumard et al. [2012]). Several studies have shown that the intensity of the SIF signal is correlated with the amount of absorbed photosynthetically active radiation (APAR) and light use efficiency (LUE) during the late morning and early afternoon (e.g. van der Tol et al. [2009]; Zarco-Tejada et al. [2013]). Other factors driving SIF are stress factors including diseases, canopy temperature, and nutrient and water availability.

Therefore, measurements of SIF are promising indicators for photosynthetic activity of plants, and thus for monitoring carbon uptake in the Earth system, for forest management, and agricultural applications.

SIF is detectable from space. Several studies have shown that SIF can be retrieved with satellite spectrometers such as MERIS [Guanter et al., 2007], GOSAT (e.g. Joiner et al. [2011]; Frankenberg et al. [2011]; Guanter et al. [2012]), GOME-2 (e.g. Joiner et al. [2013]; Köhler et al., 2015; Sanders et al. [2016]), SCIAMACHY [Khosravi et al., 2015; Joiner et al., 2016], the Orbiting Carbon Observatory 2 (OCO-2, Frankenberg et al. [2014]; Sun et al. [2018]), and recently TROPOMI [Köhler et al., 2018]. Measurements by the GOME-2 sensors are particularly interesting because of the long mission lifetime of three successive MetOp satellites (2007 up to 2025-2030) with global coverage between 1 and 2 days depending on the number of instruments in orbit and their configuration. This long data record provides good prospects to generate a climate data record of SIF (e.g. Parazoo et al., 2019). The spectral resolution of GOME-2 (~0.5 nm) allows a technique that matches the observed reflectances close to the wavelength of the far-red fluorescence peak to model reflectances. The modeled spectrum consists of contributions from surface reflectance, atmospheric transmittance, and fluorescence, where the latter fills in the Fraunhofer lines present in the incoming sunlight. This technique was pioneered by Joiner et al. [2013] and Köhler et al. [2015], and explored further by Sanders et al. [2016], who established the Sun-Induced Fluorescence of Terrestrial Ecosystems Retrieval (SIFTER) algorithm. Koren et al. [2018] used SIFTER fluorescence retrievals to study the effect of the 2015/2016 El Niño Amazon drought on the capacity of that tropical forest to store carbon relative to previous years. While the study suggested that SIFTER retrievals appropriately account for water vapour absorption signatures in the satellite spectra over the Amazon, and provide a decadal data record, there were also clear indications for spurious negative trends in SIF that need further investigation.

Here we revisit KNMI's (Royal Netherlands Meteorological Institute) SIFTER retrieval approach for GOME-2. We propose a number of improvements to the SIFTER approach based on radiative transfer modeling tests. These improvements focus on optimizing the spectral fitting window, and on calculating atmospheric transmittance terms from satellite spectra over a reference region without vegetation. The reference region is selected and sampled such that the transmittance terms are as as representative as possible (in terms of surface reflectivity, viewing geometries, atmospheric properties) for scenes with SIF. We then introduce two important quality control filters that test for the presence of unresolved spectral structures and for the presence of clouds, and address the need for a latitudinal bias correction as well as a correction for degradation in the level-1 reflectance input data. We compare the results of our new SIFTER v2 algorithm to SIFTER v1 and the data generated by NASA, and discuss the uncertainty budget and limitations.

## 2 Data and methods

### 2.1 GOME-2 sensors

The GOME-2 spectrometers onboard EUMETSAT's MetOp-A (launched 19 October 2006), MetOp-B (launched 17 September 2012) and MetOp-C (launched 7 November 2018) satellites fly in a sun-synchronous polar orbit with a local equator crossing time of 09:30 hrs local time in the descending node. For the SIF retrievals, we use (ir)radiances measured in the near-infrared channel (band 4, 593-790 nm), which has a spectral sampling of ~0.2 nm and a spectral resolution of ~0.5 nm. The scanning mirror of GOME-2 nominally covers a 1920 km swath back (1.5 sec) and forth (4.5 sec) in 6 seconds. The integration time of the detector is 0.1875 seconds resulting in 24 forward pixels with a nominal spatial resolution of $80 \times 40$ $km^2$ and 8 backward pixels at $240 \times 40 \ km^2$. Only forward-scan pixels are processed in our data product. For the nominal swath the GOME-2 instrument achieves global coverage within a day beyond 40° latitude.

The launch of GOME-2B in 2012 motivated a reduction of the GOME-2A swath in order to achieve a better spatial resolution. The GOME-2A swath was reduced on 15 July 2013 to 960 km resulting in a ground pixel size of $40 \times 40$ km$^2$ [Munro et al., 2016]. Although the detectors in the GOME-2A sensors remain the same, the narrowing of the swath brings about changes in the viewing geometries and in the field-of-view. In what follows, we therefore consider that there are
actually two GOME-2A sensors: one before 15 July 2013 (large pixels, wide range of viewing angles), and one after that date (small pixels, smaller range of viewing angles).

As other spaceborne sensors measuring reflected sunlight, GOME-2A is sensitive to degradation because of prolonged exposure to solar radiation and contamination of optical elements [Munro et al., 2016; Hassinen et al., 2016]. The exact
cause of the degradation is partially attributed to scan mirror surface contamination (also observed in GOME and SCIAMACHY) and partially due to contamination of optical surfaces in the rest of the optical light path by an unknown source of contaminants. The signal degradation is largely limited to the regions below 450 nm, but a change in reflectance values is observed throughout the spectral range, due to slightly different signal throughput variations in the solar (irradiance) optical path and the Earthshine path (because of additional degradation of the solar diffuser). This also affects
the absolute radiometric accuracy in the longer wavelength channels, like in band 4, but to a significantly lesser extent than in the short-wave channels. Since signal levels in band 4 remain very high at low noise levels, the signal-to-noise ratio is in contrast hardly affected in the region around 740 nm that are used for fluorescence retrieval. Furthermore, the level-1b data available until now have been processed with different processor versions by EUMETSAT. Until May 2014, level-1b data were reprocessed with processor version 5.3, but since then a number of processor revisions have taken place (Supplemental
Table 1).

To assess if signal throughput degradation or level-1 processing has influenced the stability and usefulness of the GOME-2A level-1b data, we track the variability of the top-of-atmosphere reflectances at 744 nm between 2007 and 2018. Figure 1 shows the time series of monthly mean reflectances (for scenes with FRESCO+ [Wang et al., 2018] cloud fraction < 0.4)
over the stable reference calibration site Libya4[1], and averaged over the globe (between 60°S and 60°N). Top-of-atmosphere reflectances are relatively stable for the early GOME-2A data with insignificant trends for the Libya4 and global mean reflectances for the period January 2007-December 2012. The late GOME-2A time series (from August 2013 onwards) however suggests a negative trend in top-of-atmosphere reflectance over Libya4 and in the global mean. The reflectance trends are consistent and amount to -0.65% yr$^{-1}$ over Libya4, and -0.71% yr$^{-1}$ averaged over the globe for August 2013 – July
2018, and we will discuss this reduction later in the context of changes in the level-1 processor versions that occurred in June 2015 (version 6.1) and January 2018 (version 6.2). We conclude that GOME-2A can safely be used for fluorescence retrieval in the early period, but for the late GOME-2A data, a degradation correction is warranted. We revisit the issue in section 4.4.

The apparent 'jump' between the baseline reflectance levels in the early (prior to July 2013) and late GOME-2A is due to the
smaller range of viewing zenith angles encountered by the late GOME-2A. When we sample only the inner 12 scan positions (VZA < 35°) for the early GOME-2A period (grey line in lower panel of Figure 1), we find a baseline reflectance level (0.196) that is consistent with the baseline reflectance level in August 2013, right after the detector settings change (0.195).

---

[1]The Libya4 site is an often-used area for vicarious calibration because of its stable surface reflecting properties. It is centered at 28.55 °N, 23.39 °E. See Neigh et al. [2015].

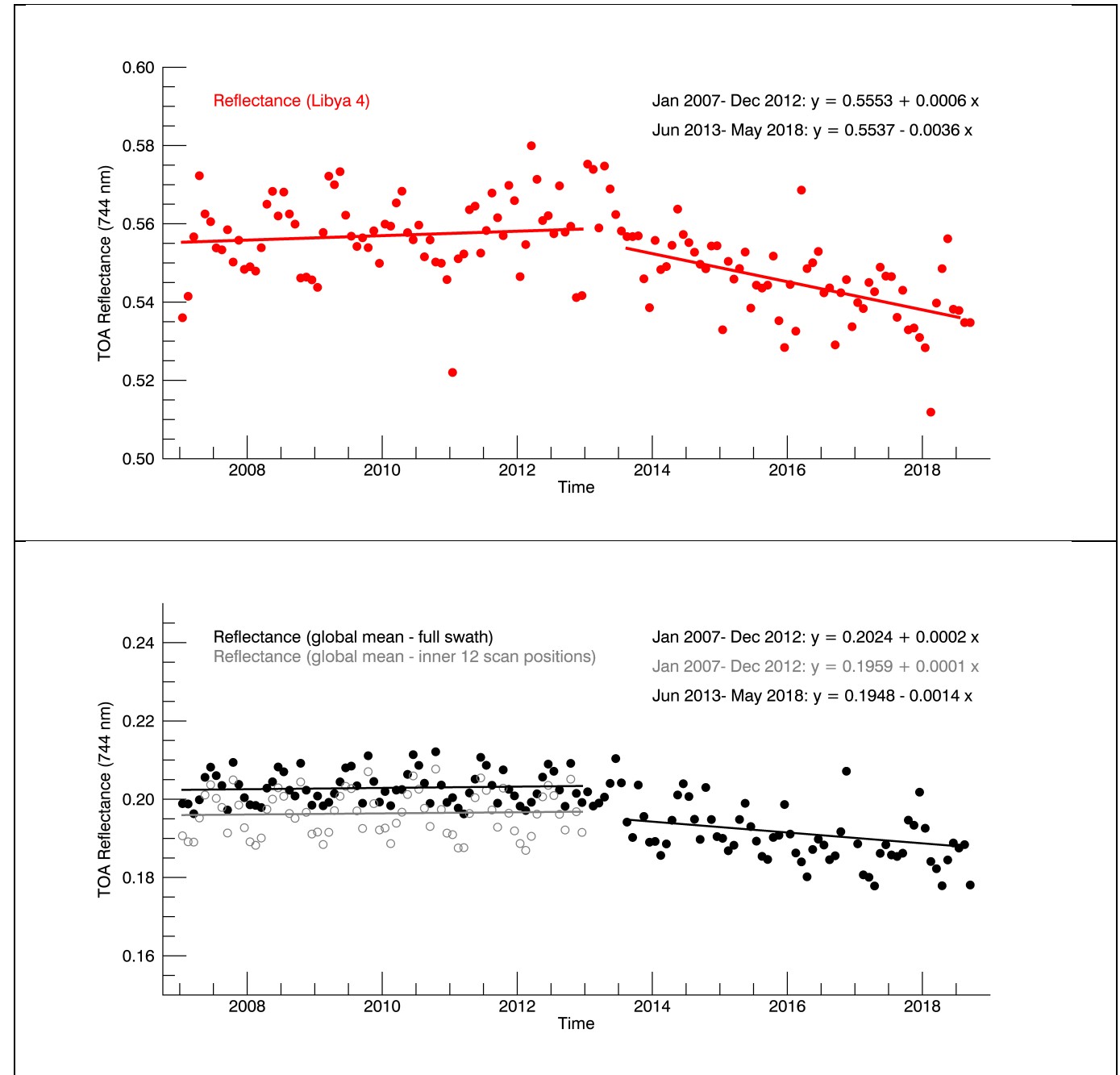

**Figure 1**. Top-of-atmosphere reflectance measured by GOME-2A at 744 nm as a function of time, for Libya4 (upper panel, red), and globally for each of the 24 scan mirror positions in the forward scan (lower panel, black), and for the 12 inner scan mirror positions (lower panel, grey). GOME-2A had 80×40 km$^2$ pixels until 15 July 2013, and a narrower swath with 40×40 km$^2$ pixels after that date. The solid lines indicate the linear trend in reflectance obtained in the early GOME-2A period for a period of exactly 6 years (January 2007-December 2012) and in the late GOME-2A period for exactly 5 years, when pixels were smaller (August 2013-July 2018). *x* in the regression equation is the fractional number of years relative to January 2007 or June 2013.

## 2.2 SIFTER retrieval algorithm

SIF can be retrieved with GOME-2 by matching a modeled spectrum that combines contributions from surface, atmosphere and fluorescence characteristics to the measured top-of-atmosphere (TOA) reflectance spectrum in a spectral region where fluorescence is known to be substantial. Under the assumption that atmospheric scattering is negligible (for a discussion see Joiner et al. [2013]), Eq. (1) describes the general model to simulate TOA reflectance (the dependency of reflectance *R* on wavelength, solar and viewing angles is omitted here for brevity) in clear-sky situations:

$$R \approx a_s T^{\uparrow} T^{\downarrow} + \frac{\pi I_{SIF} T^{\uparrow}}{\mu_0 E_0} \qquad (1)$$

with $a_s$ the surface albedo, $T^{\uparrow}$ and $T^{\downarrow}$ the upward and downward atmospheric transmission factors, $I_{SIF}$ the fluorescence
emission, $\mu_0$ the cosine of the solar zenith angle, and $E_0$ the solar irradiance at the top of the atmosphere. The surface albedo
and SIF emission are assumed to be spectrally smooth, but the transmission has distinct and variable spectral features from
solar Fraunhofer lines, as well as atmospheric oxygen and water vapour absorption signatures. Disentangling the smooth
albedo and fluorescence contributions to the TOA reflectance is possible because a number of Fraunhofer absorption lines
near the fluorescence peak at 740 nm are dampened, or 'filled in' by the additional fluorescence emission from the surface.
In the absence of fluorescence, the Fraunhofer lines in the TOA reflectance spectrum are considerably deeper because in
those circumstances the elastic scattering of sunlight at the surface maintains the relative spectral structure of the Fraunhofer
lines.

The retrieval now proceeds in two distinct steps:

(1) First, a table with possible solutions describing the atmospheric transmittance for a wide variety of viewing
conditions is constructed. A large ensemble of observed reflectance spectra is collected over a so-called reference region that
is sufficiently large and known to be free of vegetation (i.e. no fluorescence). The contribution of the surface albedo to these
reference sector TOA reflectance spectra is removed by subtracting a 2$^{nd}$ order polynomial function obtained by fitting to the
reflectance in three spectral bands with virtually no atmospheric absorption (712-713 nm, 748-757 nm and 775-783 nm). In
these bands, the TOA reflectance may be assumed to be close to the surface albedo as long as aerosol loading is modest and
clouds are absent. These conditions are fulfilled by accepting only scenes with effective cloud fractions (here from
FRESCO+) below a certain threshold (effective cloud fraction < 0.4) in the ensemble. The albedo-corrected spectra
(centered around the 740 nm far-red fluorescence peak) are then aggregated and transformed into principal components
$f_k(\lambda)$, described further in Section 4. The main components (PCs) can be interpreted as spectral features that explain the
variance in spectra devoid of surface effects, i.e. dominated by atmospheric absorption. $f_0(\lambda)$ describes the main principal
component, i.e. the mean transmission spectrum, and $f_1(\lambda)$ represents the second most important source of variability from
e.g. water vapour absorption. Higher order PCs represent variability caused by water vapour, oxygen, and from other sources
such as noise, unresolved surface and instrumental effects.. As each PC aims to maximize the explained variance of the
transmission signal, a linear combination of the first couple of components usually explains most of the transmission
signal.

(2) Then, for each individual pixel, the differences between the observed ($R_o(\lambda)$) and modeled reflectance spectrum
($R_m(\lambda)$) are minimized with a Levenberg-Marquardt least-squares regression within an appropriate spectral window. The fit
model is a representation of Eq. (1) and reads:

$$R_m(\lambda) \approx \left( \sum_{j=1}^{n} a_j \lambda^j \right) e^{-\sum_{k=1}^{m} b_k f_k(\lambda)} + \frac{\pi c I_{SIF}(\lambda)}{\mu_0 E_0} e^{-\frac{\mu^{-1}}{\mu^{-1}+\mu_0^{-1}} \sum_{k=1}^{m} b_k f_k(\lambda)} \qquad (2)$$

where $a_j$ are the fit coefficients best describing the surface reflectance contribution, $b_k$ the coefficients that best match the
set of PCs to the contribution from the transmittance, and $c$ the fitting coefficient that, multiplied with $I_{SIF}$, describes the
intensity of the fluorescence signal. As there is a total of $n+m+1$ fitting parameters ($n$ for albedo, $m$ for transmittance, and 1
for SIF) or unknowns, the spectral fitting window should be wide enough to establish an overdetermined system, i.e. include
far more than $n+m+1$ spectral samples. Leaf-level measurements over vegetated fields suggest that the SIF signal resembles

a Gaussian with a width of 34 nm and a peak at 737 nm [Daumard et al., 2012], and recent work suggests that the shape of the fluorescence spectrum is stable, especially for wavelengths larger than 740 nm [Magney et al., 2019]. A careful trade-off is required in the selection of the fitting window: it should preferably overlap with the stable part of the fluorescence reference spectrum (less uncertainty), contain several Fraunhofer lines, but preferably avoid strong absorption features from oxygen and water vapour, as will be discussed in 3.2.

In essence, the strength of the SIF signal is determined from the degree to which Fraunhofer lines are filled in. The retrieval method minimizes the differences between modeled and observed reflectance by simultaneously fitting surface albedo, atmospheric transmittance (important for wide spectral windows such as used here), and SIF. The combination of fit coefficients that best reproduces the observed reflectance spectrum, is considered to be the solution to the retrieval problem.

## 2.3 Retrieval improvements

We revisited a number of settings of the SIFTER v1 algorithm described in Sanders et al. [2016]. A motivation for our work is that the SIFTER v1 and NASA SIF products compare reasonably well on the global scale, but show little spatio-temporal correlation over the majority of the tropical forests (Figure 9 in Sanders et al. [2016]). Another incentive was the anomalous and persistent decrease in SIF over the Amazon in various studies, which required substantial detrending efforts [Koren et al., 2018] In order of relevance, these settings of interest include:

(1) the selection of the spectral fitting window,

(2) the number of principal components (PCs) used to calculate the table of transmittance, and

(3) the choice for the reference sector and calculation method for the PCs,

(4) the application of a correction for the degradation of GOME-2A radiances.

In SIFTER v1, a wide fitting window was selected (712-783 nm), which includes spectral features from the oxygen-A band (759-769 nm) and water vapour absorption (714-734 nm). These features potentially complicate the calculation of the transmittance terms with the principal component method. Here we investigate the possibility to reduce the number of PCs $f_k(\lambda)$ by selecting a narrower window that includes the strong fluorescence signature, but excludes the adjacent $O_2$-A and water vapour features.

In SIFTER v1, PCs were calculated from all top-of-atmosphere spectra taken over the non-vegetated parts of the Saharan desert in the twelve months preceding the GOME-2A measurement of interest. We test to what extent these Saharan-based PCs are representative, i.e. contain comparable atmospheric properties and viewing geometries for the retrieval conditions encountered for vegetated areas and locations.

## 3 Observation simulation experiments

### 3.1 Base test

In order to test our assumptions to model atmospheric transmittance, which is crucial for fluorescence retrieval, we simulated top-of-atmosphere spectra for a wide range of conditions with the Determining Instrument Specifications and Analyzing Methods for Atmospheric Retrieval (DISAMAR, De Haan [2011]) radiative transfer model, developed at KNMI (based on Doubling-Adding KNMI, DAK). DISAMAR generates TOA reflectance spectra, accounts for absorption by $H_2O$ and $O_2$, and describes the effects of multiple scattering and a spectrally varying surface albedo and allows convolution of a simulated reflectance signal with the pre-defined GOME-2 slit function (width ~0.5 nm). The simulations with DISAMAR allow us to control important retrieval variables such as atmospheric water vapour content, the effects of oxygen absorption, surface albedo, and viewing geometries. A top-of-canopy fluorescence signal can also be included to test which retrieval settings

(spectral fitting window, number of PCs) best reproduce the magnitude of the fluorescence under different circumstances in a so-called end-to-end test.

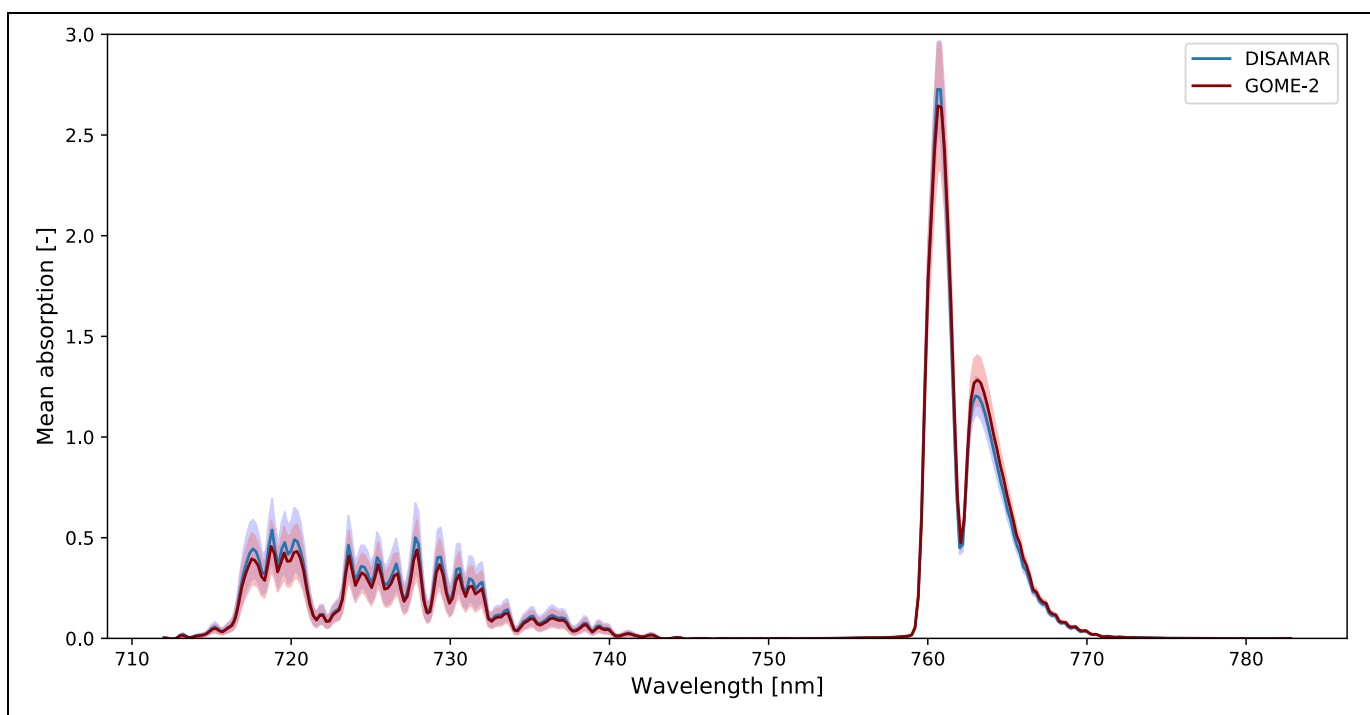

**Figure 2.** Mean wavelength-dependent absorption spectra for GOME-2A (red, $n \approx 30{,}000$) and DISAMAR-generated (blue, $n = 2{,}000$) top-of-atmosphere spectra, with $n$ the number of spectra. The mean absorption corresponds to the two way slant optical thickness resulting from water vapour and oxygen absorption. The shading shows one standard deviation.

In this first –base– test, we generated 2,000 DISAMAR TOA spectra in the range 712-783 nm for different prescribed parameters (albedo, surface pressure) and viewing geometries representative for those encountered by GOME-2. The prescribed parameters were obtained from a random draw of parameters found in the SIFTER v1 data set. In the base test there was no fluorescence, as the intention was to simulate background reference spectra to be used for the determination of the PCs. We evaluated DISAMAR TOA reflectance spectra simulated for the 'base' case against a large ensemble of
observed GOME-2A spectra over the Sahara. Figure 2 compares the mean DISAMAR and GOME-2 atmospheric absorption signatures in the 712-783 nm range. The good agreement between the two spectra provides confidence in DISAMAR as a test tool.

**3.2 Spectral fitting window experiments**

The five other experiments focused on addressing a particular aspect of the SIF retrieval (influence of water vapour absorption, albedo characteristics, and viewing geometry), and included various strengths of fluorescence. For these experiments, 1,000 TOA spectra were generated (with noise). The overall aim of the experiments was to establish the optimal spectral window and number of PCs with which to retrieve SIF. To do so, we evaluate four different spectral windows: a wide window stretching out from 712 to 783 nm, as was used in SIFTER v1, and three smaller windows that
avoid the main absorption signatures from the $O_2$-A band (712-758 nm), from water vapour (734-783 nm), and a small window avoiding both bands (734-758 nm).

Table 1 summarizes the settings used in the experiments, and their purpose. The 'Fluorescence' experiment has settings that are identical to the 'Reference' case, and allows us to investigate to what extent SIF can be reproduced in optimal conditions,
as if fluorescence would be monitored over a Sahara-type surface. The 'Water' experiment replaces the water vapour

columns over the Sahara by amounts that are more representative for tropical rain forests (ERA-Interim fields, Berrisford et al. [2011]). This allows us to assess the influence of strong water vapour absorption not fully represented in the set of PCs on the SIF retrieval. The two 'Vegetation' experiments evaluate the sensitivity of SIF retrieval to different surface albedos that are more representative for vegetated areas than in the Sahara reference region. In the last experiment ('Geometry'), the
viewing geometry is changed to angles found over a Russian boreal forest (55°-65° N) on a late summer day.

**Table 1.** Settings for the five experiments with DISAMAR. All experiments were done without clouds, and signal-to-noise ratio of the simulated reflectance spectra was 1,000. SZA stands for solar zenith angle, and VZA for viewing zenith angle.

| Experiment | SZA range | VZA range | $H_2O$ column (kg $m^{-2}$) | Surface albedo | Fluorescence (mW $m^{-2}$ $sr^{-1}$ $nm^{-1}$) | Purpose |
|---|---|---|---|---|---|---|
| Fluorescence | 21.4°-66.8° | 0.0°-53.8° | 4.0-40.0 | 0.41-0.45 | 0.0-4.0 | Reproduce SIF in ideal conditions with noise |
| Water | 21.4°-66.8° | 0.0°-53.8° | 30.0-65.0 | 0.41-0.45 | 0.0-4.0 | Test influence of too low $H_2O$ in PCs and SIF retrieval |
| Vegetation Albedo | 21.4°-66.8° | 0.0°-53.8° | 4.0-40.0 | 0.4 | 0.0-4.0 | Test influence of constant spectral albedo in PCs and SIF retrieval |
| Vegetation Red Edge | 21.4°-66.8° | 0.0°-53.8° | 4.0-40.0 | 0.06-0.5[2] | 0.0-4.0 | Test influence of realistic red edge spectral albedo in PCs and SIF retrieval |
| Viewing Geometry | 54.4°-69.6° | 0.0°-53.8° | 4.0-40.0 | 0.41-0.45 | 0.0-4.0 | Test whether SIF can still be retrieved for viewing conditions at high latitudes |

In the (Saharan) 'Fluorescence' experiment, we tested to what extent the fluorescence assumed in the generation of the TOA spectra could be reproduced with the retrieval approach described in Section 2. Table 2 summarizes the results of this experiment in terms of retrieval bias and uncertainty. The results from this end-to-end test strongly suggest that limiting the spectral fitting window, and thereby minimizing the number of PCs required, leads to the most accurate reproduction of the fluorescence signal. Clearly, the combination of a narrow fitting window (734-758 nm) and a low number of PCs (8) gives
an unbiased retrieval and lowest uncertainties (approximately 25%).

**Table 2.** Results of the 'Fluorescence' experiment (mean of 1,000 spectra) to reproduce fluorescence for different fitting windows and number of PCs used. The bias is defined here as the mean of the differences between assumed and retrieved fluorescence strength (on average was 1.5 mW $m^{-2}$ $sr^{-1}$ $nm^{-1}$), and the RMSE stands for the root of the mean of the squared
deviations. Faulty retrievals were not included in the calculation of the bias or the RMSE. 'Faulty' retrievals are characterized by high spectral autocorrelation (> 0.2) in their fit residuals. All experiments were done without clouds, and signal-to-noise ratio of the simulated reflectance spectra was 1,000.

| | Bias (mW $m^{-2}$ $sr^{-1}$ $nm^{-1}$) | RMSE (mW $m^{-2}$ $sr^{-1}$ $nm^{-1}$) | Faulty |
|---|---|---|---|

---

[2]The spectral albedo increases from 0.06 at 710 nm to 0.5 at 750 nm (see Figure 2.2 in van Schaik [2016]).

| | | | |
|---|---|---|---|
| 712-783 nm (SIFTER v1) 8 PCs | -0.47 (-31%) | 0.71 | 34.7% |
| 20 PCs | -0.37 (-25%) | 0.59 | 30.6% |
| 35 PCs | -0.37 (-25%) | 0.62 | 28.8% |
| 712-758 nm (exclude O$_2$-A band, keep H$_2$O) 8 PCs | -0.20 (-13%) | 0.46 | 23.6% |
| 20 PCs | -0.23 (-15%) | 0.50 | 20.1% |
| 35 PCs | -0.26 (-17%) | 0.53 | 18.2% |
| 734-783 nm (exclude H$_2$O band, keep O$_2$-A) 8 PCs | -0.49 (-33%) | 0.57 | 41.9% |
| 20 PCs | -0.48 (-32%) | 6.41 | 37.9% |
| 35 PCs | -0.25 (-17%) | 0.49 | 35.4% |
| 734-758 nm (exclude both bands) 8 PCs | 0.0 (0%) | 0.39 | 16.5% |
| 20 PCs | 0.04 (3%) | 0.43 | 16.2% |
| 35 PCs | -0.03 (-2%) | 0.5 | 14.6% |

As a further test of the retrieval, we evaluated the goodness-of-fit, using the concept of spectral autocorrelation (Supplement Section 2). For every spectrum, we analyze the residuals of the fit remaining after minimizing the differences between the DISAMAR spectrum and the modeled spectrum from Eq. (1). In general, high fit residual values are considered to be indicative of a poor fit, but the degree of spectral lag-one autocorrelation in the residuals is another indicator as it tests the non-randomness of the residuals [NIST/SEMATECH, 2018]. If the modeled spectrum perfectly explains the TOA spectrum, fit residuals appear as noise with zero autocorrelation. But if the model fails to resolve structural components of the TOA spectrum, then this will show up as structure in the fitting residuals, and, consequently, autocorrelation will be higher than zero. Our tests suggest a strong relationship between the bias and the value for $\chi^2_{red}$ in the retrieval and the degree of autocorrelation in the residual spectrum, when the autocorrelation exceeds 0.2 (See Figure S2). This relationship is apparent for all selections of fitting window and number of PCs. We consider retrievals 'faulty' when the spectral autocorrelation in the fit residuals exceeds 0.2 (corresponding with $\chi^2_{red} > 3$), also for fluorescence retrievals from GOME-2. Auto-correlation is strongly correlated with the fitting residuals (or RMS error), and both metrics are included in the data file. For non-faulty retrievals in Table 2, the mean $\chi^2_{red}$ value was close to 1, e.g. 1.23 and 0.92 for the 734-758 nm window with 8 and 35 PCs respectively, and 1.49 and 1.14 for the 734-783 nm window with 8 and 35 PCs.

As to *why* including the O$_2$-A band harms the retrieval of SIF from space, we performed an additional study into the sensitivity of top-of-atmosphere radiance to SIF at the surface. We simulated TOA radiances for two ensembles: one without SIF and one with a SIF strength of 4.0 mW m$^{-2}$ nm$^{-1}$ sr$^{-1}$ (at 737 nm). The settings in DISAMAR were such that the ensemble average surface albedo, surface pressure, and viewing geometry were the same, so the essential difference between the two ensembles is in the presence of a SIF signal. No clouds or aerosols were included in the simulations.

The DISAMAR simulations show that the presence of a SIF signal leads to a small addition of radiance across the spectrum (upper panel Figure 3). The surplus radiance closely follows the magnitude and spectral shape of the fluorescence source spectrum between 740-758 nm, but is weaker between 734-740 nm and 759-766 nm, where water vapour and oxygen partly absorb the SIF signal travelling from the Earth's surface towards the sensor. The sensitivity of the radiances to changes in the 'state' thus shows a strong spectral dependence. Put simply, within the $O_2$-A band transmission is low and only half of the SIF signal makes it to the sensor. But between 740 nm and 758 nm (and also for 768-783 nm), transmission is full and all SIF photons reach the sensor.

To quantify this non-uniform sensitivity to SIF across the spectral range, we may use the concept of the air mass factor (AMF), which is widely used for the retrieval of weakly absorbing gases in the atmosphere. The AMF for an absorber at a vertical layer $l$ can be calculated as $M_l(\lambda) = \frac{\partial I(\lambda)}{\partial \tau_l(\lambda)}$ (e.g. Eskes and Boersma [2003]), where $\partial I(\lambda)$ is the change in TOA radiance caused by the addition of optical thickness $\partial \tau_l(\lambda)$ by an absorber in layer $l$. Scattering of light by air molecules and the generally low surface albedo results in small UV-Vis air mass factors, i.e. low vertical sensitivities for trace gases close to the Earth's surface.

In the near-IR, atmospheric scattering is an order of magnitude weaker than in the UV-Vis, so that transmission is generally close to 100% (very long light paths being the exception). TOA sensitivity to SIF at the Earth's surface is anticipated to be optimal. To evaluate the spectral sensitivity to fluorescence, we define a spectrally resolved SIF AMF as $M_{SIF}(\lambda) = \frac{\partial I(\lambda)}{\partial F(\lambda)}$, with $\partial I(\lambda)$ the change (here: addition) in the TOA radiance caused by the addition $\partial F(\lambda)$ of fluorescence at the Earth's surface. In the 'ideal' retrieval situation, the AMF has a value of 1, i.e. full sensitivity to SIF at the surface. In sub-optimal conditions, such as when optically thick clouds hide the SIF from detection, the AMF will be zero (e.g. Frankenberg and Berry [2018]). Within absorption bands, the AMF may be lower than 1, indicating reduced sensitivity to SIF.

The middle panel of Figure 3 shows that the SIF AMFs calculated from the DISAMAR between 742 and 758 nm are indeed close to 1, demonstrating the good sensitivity to fluorescence in this spectral range for our ensemble. Between 734-742 nm the AMF has values close to 0.9, and within the $O_2$-A absorption band the AMF drops to values of ~0.5. This explains why a spectral fit with a wide spectral window that includes the $O_2$-A band will not reproduce but rather underestimate the SIF signal prescribed in the simulations. The spectral fitting procedure attempts to match all spectral features within the window. For the wide window this comprises the in-filling of the Fraunhofer lines in spectral regions where sensitivity to SIF is close to 1, but also where sensitivity to SIF drops to 0.5, i.e. the in-filling within the $O_2$-A band. The single, 'window-average' retrieved SIF value then becomes a trade-off between partial SIF in-filling within the $O_2$-A band and complete in-filling outside the absorption bands. The result is a compromise, a structural underestimate of SIF. The lower panel of Figure 3 shows the diagnosis: we observe much larger average spectral residuals (model minus observation) for the 734-783 nm window in grey, especially around the Fraunhofer features, than for the 734-758 nm window (in black).

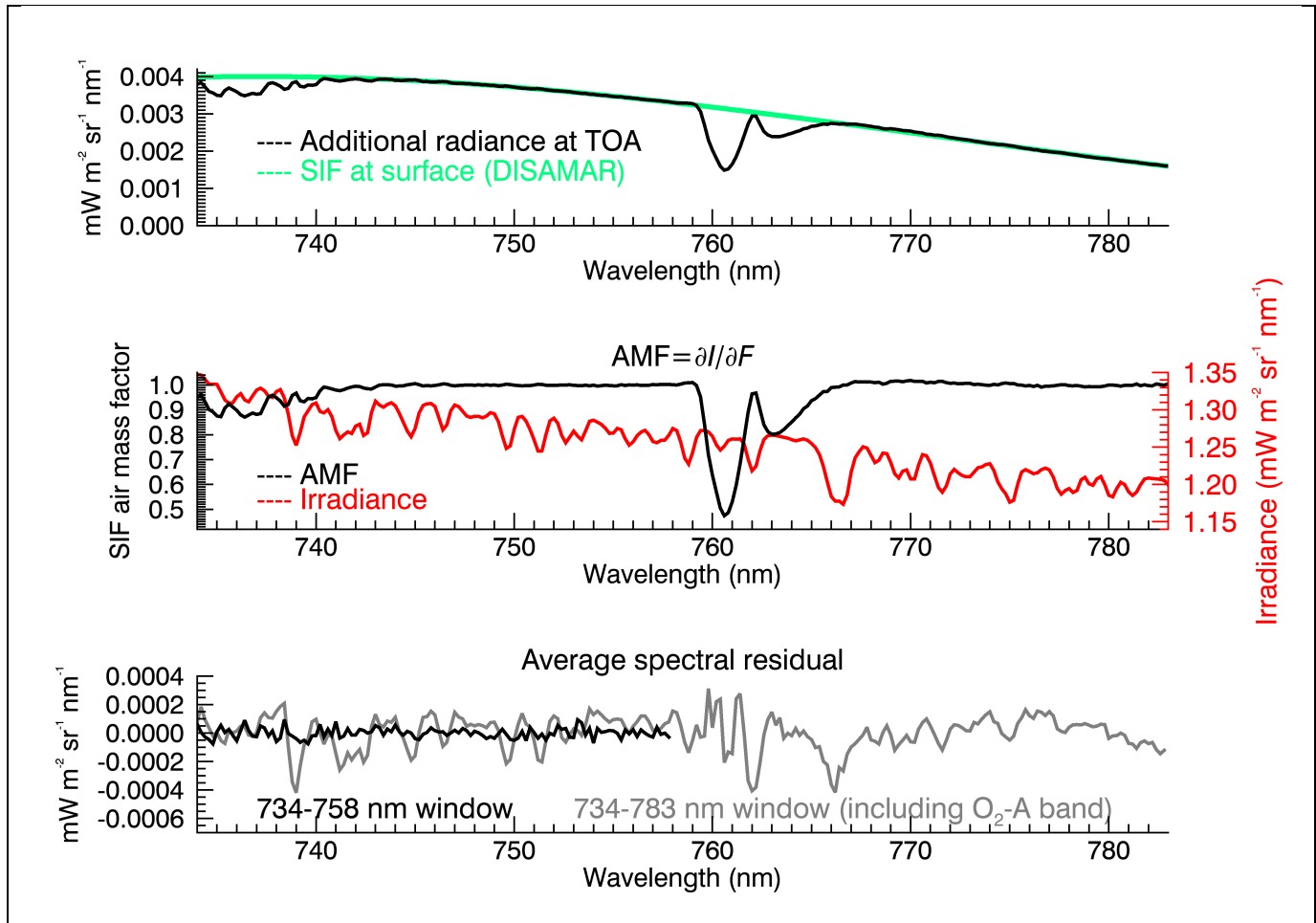

**Figure 3.** Upper panel: Difference between ensemble average simulations of top-of-atmosphere radiances with and without a SIF signal present at the surface (black line). The DISAMAR ensemble consists of 140 simulations with an average surface pressure of 977 hPa, surface albedo of 0.46 (at 750 nm), and geometrical AMF (defined as $(\cos \theta)^{-1}$) of 1.21. The fluorescence at the surface in green has a Gaussian shape with a peak at 737 nm of 4 mW m$^{-2}$ nm$^{-1}$ sr$^{-1}$ and a width of 33.9 nm. Middle panel: SIF AMF as a function of wavelength, and solar irradiance (red) based on the high-resolution spectrum from Chance and Kurucz convolved with the GOME-2A slit function. Lower panel: average spectral fitting residuals for end-to-end retrievals with the narrow (734-758 nm) and wide (734-783 nm) windows.

In the 'Water vapour experiment', we investigate which fitting window is optimal when higher water vapour concentrations (30-65 g m$^{-2}$) than encountered over the Sahara reference region (4-40 g m$^{-2}$) are encountered. The detailed results are summarized in Table S2. Again, the retrieval in the narrow fitting window (734-758 nm) performs best in reproducing the
prior fluorescence levels (bias <10%), albeit with a considerable fraction of retrievals with autocorrelation > 0.2 (40-65%). Retrievals with other fitting windows all performed worse in reproducing the fluorescence signal, reflecting that the relatively strong water vapour signatures around 730 nm lead to more unexplained structure in the fit residuals. The other three experiments ('vegetation albedo', 'vegetation red edge', and 'viewing geometry') led to very similar conclusions: fluorescence is best reproduced with a 734-758 nm window, and with 8 PCs. Inclusion of a distinct 'red edge' in the spectral
surface albedo (700-730 nm) did not affect the small fitting window very much, whereas fits with the wider fitting windows suffered to reproduce the assumed fluorescence levels. One notable result is that changing the viewing geometries to those representative of a late-summer boreal morning over Russia, did not strongly affect the retrieval results either. This suggests that a reference set based on the relatively small range of solar zenith angles encountered over the Saharan reference region can still be used for retrieving fluorescence over high-latitude regions. Table 3 presents the overview of the main results for
the narrow fitting window.

To summarize, our tests show that fluorescence is consistently reproduced with a narrow fitting window (734-758 nm) and 8 PCs under a wide range of different, but realistic simulated retrieval conditions. This 734-758 nm window is similar to the fitting window selected by Joiner et al. [2016]. The test results also suggest that it is important to filter out retrievals with persistent spectral structure in their residuals, and this can be done via the auto-correlation test proposed here. Higher amounts of atmospheric water vapour than present over the Sahara may lead to absorption imprints in the TOA spectra not captured by the PCs, and this leads to a relatively large fraction of faulty retrievals. This indicates that defining a reference sector to include scenes with similar water vapour amounts as encountered over vegetated areas has good potential to improve the retrieval. We will investigate this further in Section 4. Limiting the spectral range of the fitting window by excluding absorption bands is effective in reducing the number of PCs needed, both for GOME-2A and DISAMAR spectra. Including the bands with $O_2$-A and water vapour absorption signatures (as in SIFTER v1) strongly reduces the explained variance of principal components 1 to 4. Wider spectral fitting windows thus require more PCs to capture atmospheric effects, and selecting a narrow fitting window in the 734-758 nm range improves our SIFTER retrievals. Our AMF analysis and Figure 3 suggest that the influence of $H_2O$ variability could be reduced further in future algorithm improvements by narrowing down the spectral window even further to 740-758 nm.

**Table 3.** Overview of the 5 experiments (mean of 1,000 spectra) to reproduce fluorescence for the narrow fitting window (734-758 nm) and 8 PCs. Bias is the mean of the differences between assumed and retrieved fluorescence strength (on average 1.5 mW m$^{-2}$ sr$^{-1}$ nm$^{-1}$), and the RMSE stands for the root of the mean of the squared deviations. Faulty retrievals were not included in the calculation of the bias or the RMSE. 'Faulty' retrievals are characterized by high spectral autocorrelation (> 0.2) in their fit residuals.

|                     | Bias (mW m$^{-2}$ sr$^{-1}$ nm$^{-1}$) | RMSE (mW m$^{-2}$ sr$^{-1}$ nm$^{-1}$) | Faulty |
|---------------------|------------------|------------------|--------|
| Fluorescence        | 0.0 (0%)         | 0.39             | 16.5%  |
| Water Vapour        | +0.12 (+8%)      | 0.42             | 64.5%  |
| Vegetation Albedo   | -0.02 (-1.5%)    | 0.44             | 13.6%  |
| Vegetation Red Edge | -0.01 (-1%)      | 0.39             | 19.1%  |
| Viewing Geometry    | +0.02 (+1.5%)    | 0.35             | 23.2%  |

## 4 Retrieval method

### 4.1 Selection reference sector and period

Two questions for the calculation of the principal components need to be addressed:

- is the pool of spectra large enough to generate a sufficiently accurate set of PCs (length of period and size of reference area)?
- are the PCs based on atmospheric scenarios sufficiently representative for vegetated areas (spatial representativeness)?

In SIFTER v1, PCs were determined from spectra observed over the barren Sahara in the previous year. The drawback of taking an ensemble from the previous year is that atmospheric conditions over the reference area may differ from year-to-year, making it difficult to compare year-to-year changes in fluorescence strength. Experiments with shorter reference periods (not shown) showed much stronger inter-annual variability in retrieved SIF, suggesting sensitivity to the period chosen. A possible explanation for this sensitivity is that variability in water vapour over the Saharan reference sector translates into variability in SIF over e.g. the Amazon region. We thus generate PCs for GOME-2A over the Saharan reference sector for the entire period 2007-2012 instead of the moving yearly window. PCs from a multi-year ensemble of

Saharan measurements capture the wider dynamic range of water vapour (as shown in ECMWF ERA-Interim data [Dee et al., 2011], and evaluated in e.g. Grossi e a. [2015]) encountered over the six-year period compared to water vapour variability within one year. GOME-2A settings changed in July 2013, so for retrievals done after that date (with a smaller field-of-view and smaller range of viewing zenith angles) the set of PCs was based on narrow-swath only measurements over the reference sector (all PCs except the last (PC9) were spectrally similar). The Sahara reference sector here is defined as the region (8°W – 29°E; 16-30°N), with scenes free of vegetation according to USGS Global Land Cover Characterization database version 2 (https://lta.cr.usgs.gov/GLCC) and effective cloud fraction [Wang et al., 2008] from FRESCO < 0.4.

We then evaluated whether the SIFTER v1 reference area over the Sahara was sufficiently representative in terms of total water vapour. Total water vapour is typically lower over deserts than over moist tropical forests. Our sensitivity tests indicate that non-linear effects due to water vapour saturation cause effects that are not captured by the PCs, and the autocorrelation filter then automatically rejects those cases (Table 3). We tested whether using spectra from a different reference sector, the moister tropical Atlantic Ocean (15-30°N, 30-45°W), was useful to generate a more representative set of PCs. Because the low ocean albedo leads to low TOA reflectance levels, we also included ocean scenes with effective cloud fractions larger than 0.4 in the ensemble. We found that the tropical Atlantic ensemble has very similar PCs (the first four were practically indistinguishable) as the Sahara set, and using Atlantic Ocean PCs instead of Sahara PCs leads to SIF values that are highly consistent with the ones retrieved with the Sahara set, consistent with Sanders et al. [2016]. This might seem counter-intuitive, as higher water vapour concentrations are present over the Atlantic Ocean than over the Sahara. However, using mostly cloudy scenes implies that the lower atmosphere is partly obscured from view. Water vapour in the lower troposphere may therefore (still) be under-represented in the TOA reflectance, and therefore in the PCs. Because using the Atlantic reference area (cloudy pixels) did not result in major differences in PCs or SIF, we keep the Sahara reference sector (for more details we refer to the Section 3.3.2 in Van Schaik [2016] and Sanders et al. [2016]).

## 4.2 Retrieval settings and demonstration

Table 4 summarizes the main changes between SIFTER v1 and SIFTER v2, as motivated by the observation simulation experiments (section 3) and tests with real data (section 4) discussed above.

As a demonstration of the retrieval principle, we show here the operation of the fitting approach from the GOME-2A spectra measured on a clear-sky Summer day (15 July 2007) over the eastern United States. We performed two different retrievals on the GOME-2A spectra: one in which the fitting model (Eq. (2)) accounted for a SIF-term, and one in which the SIF-term was omitted (a reflectance model with only albedo and transmission contributions on the right-hand side of Eq. (2)). Figure 4 shows the observed and modeled radiances for the two approaches over the eastern United States on 15 July 2007. The differences between the two modeled and observed radiances are indistinguishable by eye, but the magnitude of the unexplained fitting residuals clearly reveals a better fit when including SIF in the fitting model. In fact, the RMS of the residuals is some 40% smaller for the fit including the fluorescence term. The residuals improve in spectral regions with prominent solar Fraunhofer lines, highlighting that the broad fluorescence signal from vegetation leads to infilling of the solar Fraunhofer lines from below, and is well detectable even for an instrument with a relatively coarse spectral resolution of 0.5 nm such as GOME-2A.

**Table 4.** Summary of individual changes in fluorescence retrievals from SIFTER v1 [Sanders et al., 2016] to SIFTER v2 (this work).

|  | SIFTER v1 | SIFTER v2 (Jan 2007-June 2013) | SIFTER v2 (Aug 2013-Dec 2018) |
|---|---|---|---|
| Spectral fitting window | 712-783 nm | 734-758 nm | 734-758 nm |
| Reference Sector region | Sahara | Sahara | Sahara |
| Number of PCs | 35 | 10 | 10 |
| Reference period | Past 12 months | Jan 2007 - Dec 2012 | Jan 2007 - Dec 2012, but only scenes with VZA<35°. |
| Albedo retrieval | 4th order polynomial (5 terms) | 4th order polynomial (5 terms) | 4th order polynomial (5 terms) |
| Solar irradiance spectrum | Chance and Kurucz [2010] spectrum, convolved with GOME-2 slitfunction and scaled with Earth-Sun distance. | Chance and Kurucz [2010] spectrum, convolved with GOME-2 slitfunction and scaled with Earth-Sun distance. | Chance and Kurucz [2010] spectrum, convolved with GOME-2 slitfunction and scaled with Earth-Sun distance. |

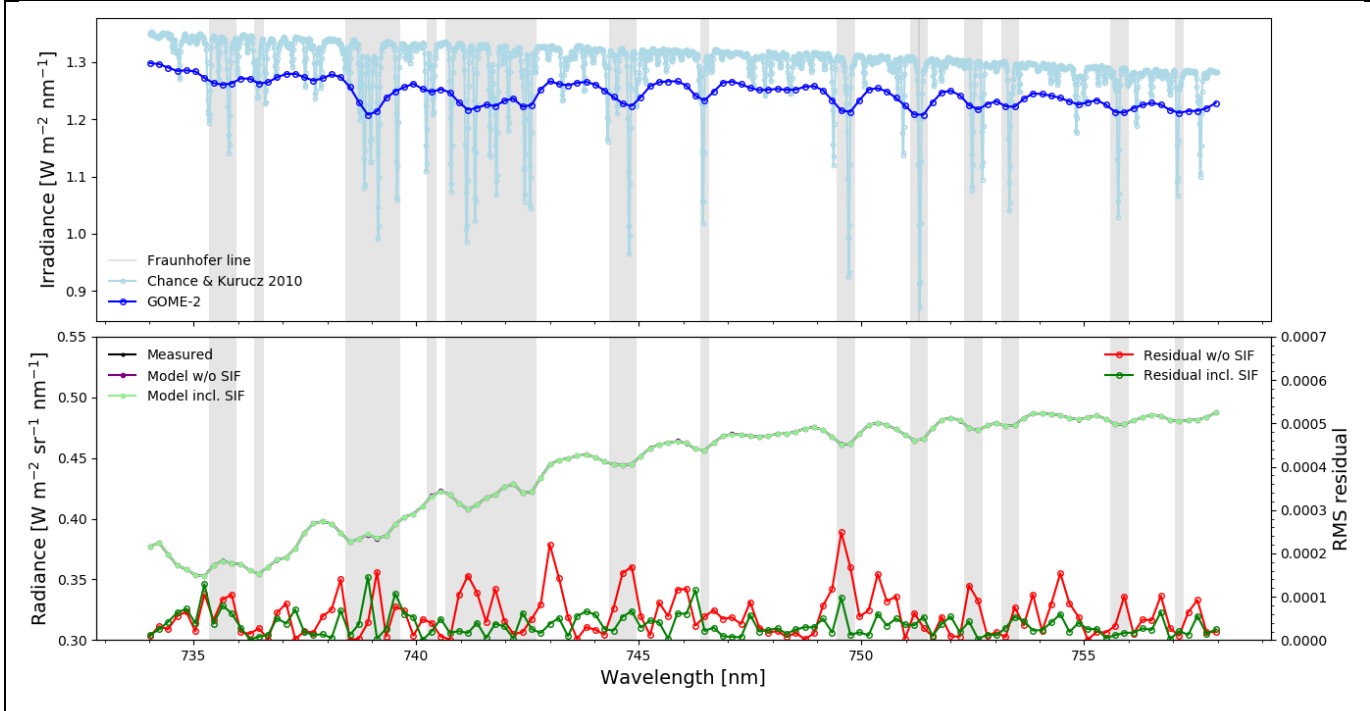

**Figure 4.** The lower panel shows the measured and modeled (with and without accounting for fluorescence) radiance spectrum in the SIFTER v2 fit window (734-758 nm) on 15 July 2007 over vegetated areas in the eastern United States (orbit 3821). The spectra represent the average over 378 cloud-free pixels with a fluorescence signal strength > 1.5 mW m$^{-2}$ sr$^{-1}$ nm$^{-1}$. The measured and modeled spectra are indistinguishable, so that only the light green line is visible. The dark green and red lines show the absolute radiance residuals for a fit including and excluding fluorescence. The upper panel shows in light blue the 0.01 nm resolution solar irradiance spectrum [Chance and Kurucz, 2010] and in dark blue the irradiance spectrum convolved with the GOME-2A slit function. The grey lines connecting the lower and upper panel indicate the spectral regions where solar Fraunhofer line infilling by fluorescence is relevant.

## 4.3 Zero-level adjustment

Similar to other retrievals (Köhler et al. [2015]; Joiner et al. [2016]), global SIFTER v2.0 retrievals show indications for a bias in the retrieved fluorescence levels that systematically depend on latitude. The cause of the bias is likely related to changes in the instrument slit function, driven by changing optical bench temperatures over the orbit [Munro et al., 2016], and resulting in an increase in slit function width along the orbit [Azam and Richter, 2015]. Our PCs capture the slit function variability due to seasonal and long-term optical bench temperature changes only as far as they occur within the reference sector over the years 2007-2012, i.e. between 0-20°N. A widening of the slit function over the orbit would have implications for the observed depth of the Fraunhofer lines: the wider the slit, the less deep the Fraunhofer lines. Shallower Fraunhofer lines are interpreted by the fitting algorithm to have been caused by fluorescence, which should lead to a positive fluorescence bias in the southern hemisphere. Conversely, sharper and deeper Fraunhofer lines (relative to the width of the lines over the Sahara), are interpreted as caused by negative fluorescence, and this is anticipated as a negative bias for latitudes north of the Sahara.

Here we make an estimate of the zero-level bias by using the fluorescence signal strength observed over the Pacific Ocean (130-150° W), where far-red fluorescence is supposed to be zero. We obtain a daily bias estimate in steps of 1° latitude, and then subtract this value for every fluorescence retrieval within the same 1°-latitude band on that day, thus assuming that the bias does not depend on longitude. The bias is determined by a linear regression of the retrieved fluorescence to the reflectance levels for all GOME-2 retrievals with a cloud fraction < 0.4 and auto correlation < 0.2 within the 1° by 20° Pacific box. The regression coefficients $a$ (offset) and $b$ (slope) then describe the bias $B_i$ for a particular pixel $i$ as $B_i = a + b \cdot R_i^{744}$ with $R_i^{744}$ the reflectance level at 744 nm for pixel $i$. Tests indicate that such bias correction results in a smoother bias estimate than simply prescribing the correction $B_i$ determined from the mean reference sector reflectance (right panel of Figure S3).

Daily bias corrections are similar to monthly mean corrections, as shown by the similarity between the daily zero-level estimate for 15 July and the average zero-level adjustment for the month of July in Fig. 4(a). Note that there are very few useful observations (due to clouds) between 40-55°N in the Pacific Ocean reference zone (130-150° W), so that on 15 July the zero-level adjustment has quite pronounced values. Figures 5(b) and 5(c) display the unadjusted and adjusted fluorescence levels retrieved on 15 July 2007. In the operational retrieval, the bias correction is determined based on the most recent days as soon as a sufficient number of pixels has been collected in a latitude bin (at least 10 per bin). In practice, the bias correction is often based on the last day, but for some latitude bands, the correction can be based on pixels dating back at most 14 days. We see only small differences over land, and the most distinct difference is the removal of the positive values over the oceans in the Southern Hemisphere. Figure S3 shows the annual mean bias correction as a function of latitude for different years. The magnitude of the bias can exceed 0.2 mW m$^{-2}$ sr$^{-1}$ nm$^{-1}$, some 10% of the highest fluorescence signals. The bias correction follows a similar pattern from year to year (Figure S3) and we do see indications for a weak long-term trend in the bias correction, most likely caused by the gradual change in optical bench temperatures, and hence in the slit function.

Over desert and high-altitude regions, negative fluorescence values are retrieved, and these may be the consequence of the Sahara reference set not being sufficiently representative for these areas. Radiative transfer tests described in Botia [2017] show that the TOA reflectance decreases with decreasing surface pressure. This suggests that observed reflectances are evaluated with too high reflectances derived from PCs over the Sahara (surface pressures 900-1000 hPa), and these

reflectance overestimations are then compensated for by negative fluorescence. This negative bias can be resolved by using PCs obtained from a reference set over barren high-altitude regions [Botia, 2017].

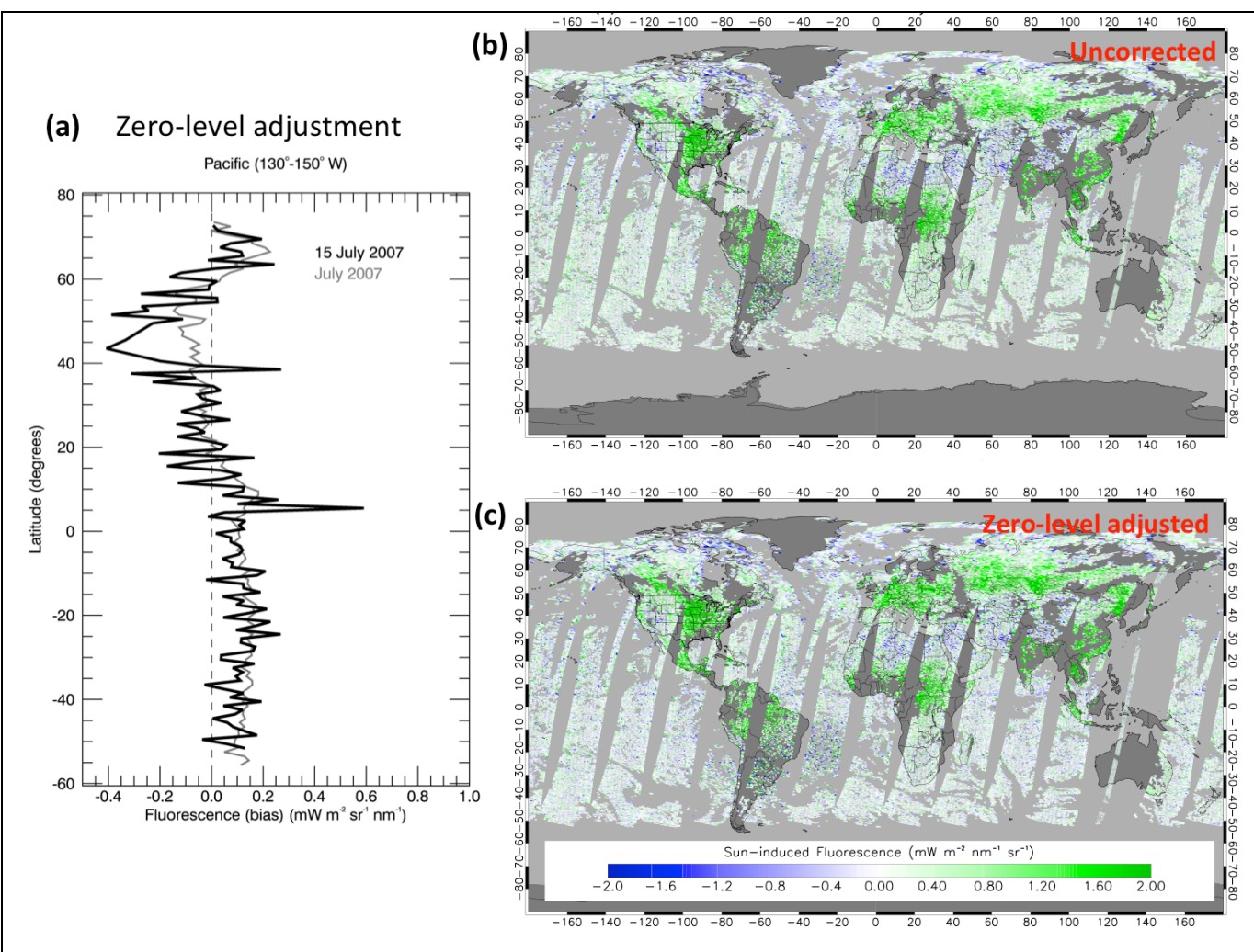

**Figure 5.** (a) Zero-level fluorescence found over the Pacific (130-150°W) on 15 July 2007 and averaged over July 2007, (b) level-2 SIF data on 15 July 2007 without zero-level adjustment, and (c) level-2 SIF data after the zero-level adjustment for 15 July 2007. The effect of the adjustment can be seen over the Southern Oceans, which after adjustment have SIF values close to 0 mW m$^{-2}$ nm$^{-1}$ sr$^{-1}$.

### 4.4 Correction for degradation in GOME-2A reflectances

Previous results with SIFTER v2, such as discussed in Koren et al. [2018], showed a substantial reduction of the SIF signal in the later years of the GOME-2A period, which required detrending. The reduction in SIF is not real but the consequence of reductions in throughput levels ("degradation") or changes in calibration key data. Here we investigate the feasibility of a correction for the reduction of the reflectances shown in Figure 1. We focus on changes in the post-2013 reflectances over the Sahara, since the principal components $f_k(\lambda)$ are derived from stable reflectances obtained over this region in the 2007-2012 period. Figure 6(a) shows the degradation of the reflectances averaged over the period that a level-1 processor was applied. The degradation is weakest in version 6.0 (June 2014-June 2015), and strongest in the v6.2 (January-December 2018) level-1 processor. The degradation changes with season (not shown). Based on these findings, we calculate

degradation correction spectra[3] for all seasons since June 2014 to account for the fact that the principal components based on 2007-2012 spectra would lead to too strong transmission terms in Eqs. (1) and (2). As an example we show correction spectra for December-January-February (DJF) in the years after the processors changed in Figure 6(b), which illustrates that the corrections generally have a spectral pattern with slightly stronger corrections required at the shortwave part of the spectrum where water vapour absorbs. The correction spectra, with values in the range from 1.00 to 1.05 are applied to the observed reflectance spectra $R_o(\lambda)$. The correction ensures that prior to fitting the observed reflectance spectrum is elevated to a level anticipated if no degradation had taken place after June 2014, so that the spectral fitting with the principal components obtained for the stable 2007-2012 period (when reflectances were higher) is still appropriate. The effect of correcting for throughput reductions on retrieved SIF levels is discussed in Section 5.3.

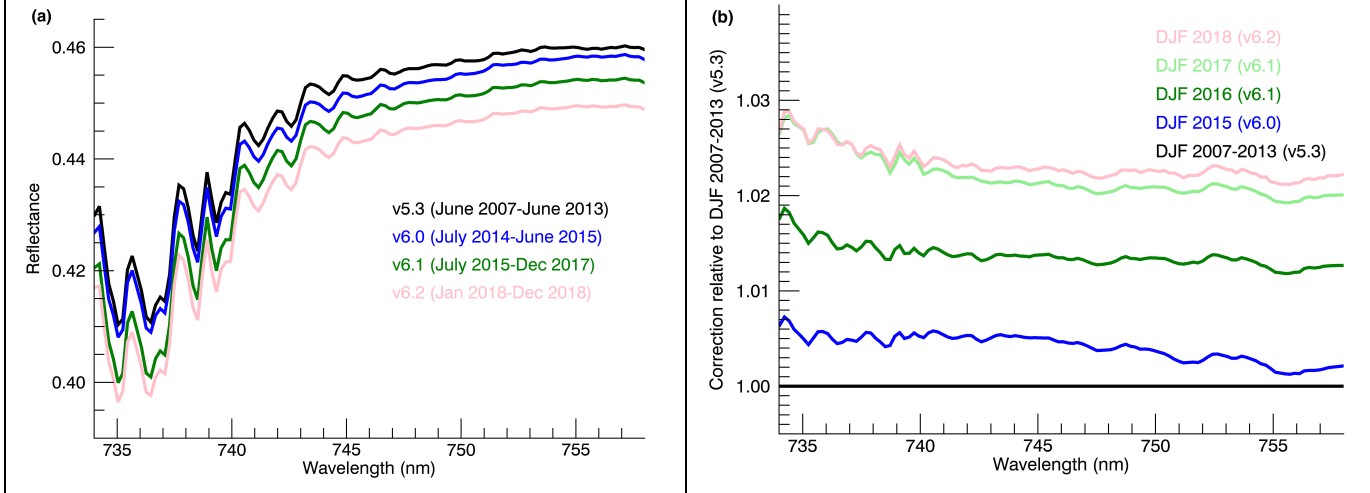

**Figure 6.** (a) Average reflectance spectra over the Sahara reference area for the different processor versions of GOME-2A. For version 5.3, only scenes with viewing zenith angles < 35° have been selected to stay consistent with the viewing geometries after the detector switch in June 2013. (b) Spectral corrections calculated as in the footnote for the winter seasons following the level-1 processor version changes.

### 4.5 Uncertainty budget

Uncertainty in the retrieved fluorescence is driven by uncertainties in the measured reflectance (i.e. by measurement noise) and by uncertainties associated with solving the inverse non-linear problem. In the non-linear Levenberg-Marquardt regression, the differences between the measured and modelled reflectance are minimized by the parameters best describing the surface albedo (5 fit parameters), the atmospheric transmission (10 parameters for the 10 principal components), and the intensity of the fluorescence via Eq. (1). Correlations among the parameters may play an important role in the fitting, and contribute to (additional) uncertainty in the fluorescence. The diagonal element of the error covariance matrix provides the uncertainty in the fluorescence fit parameter. We can consider this a realistic estimate of the uncertainty for a single retrieval, as long as the description of surface albedo and transmission is appropriate, and does not suffer from possible misrepresentations. Such misrepresentations could be too low extinction from atmospheric water vapour described in the set of PCs (as compared to the scene of interest), or too much extinction in the PCs for scenes with higher terrain height than the Sahara [Botia, 2017].

---

[3] The seasonal correction spectra for e.g. version 6.0 (July 2014-June 2015) are defined as $c(\lambda) = \frac{R_{S,v5.3}(\lambda)}{R_{S,v6.0}(\lambda)}$, i.e. the ratio between the mean climatological (2007-2012) reflectance spectrum in a particular season over the Sahara in version 5.3, and that of the seasonal reflectance spectrum over the Sahara in version 6.0.

Figure 7 shows typical pixel uncertainties obtained from the Levenberg-Marquardt fit as a function of the fluorescence signal strength over land (left panel), and as a function of scene brightness (right panel). The median single-pixel uncertainty is 0.6 mW m$^{-2}$ sr$^{-1}$ nm$^{-1}$, and is independent of the fluorescence signal itself. This implies that the relative uncertainty for strongly fluorescent scenes is 25%-50%, but can easily exceed 100% for scenes with a moderate signal of <0.5 mW m$^{-2}$ sr$^{-1}$ nm$^{-1}$. The right panel shows that fluorescence retrieval is most precise for scenes that are relatively dark. This illustrates that the additional top-of-atmosphere radiance signal from SIF is optimally retrieved against a dark background.

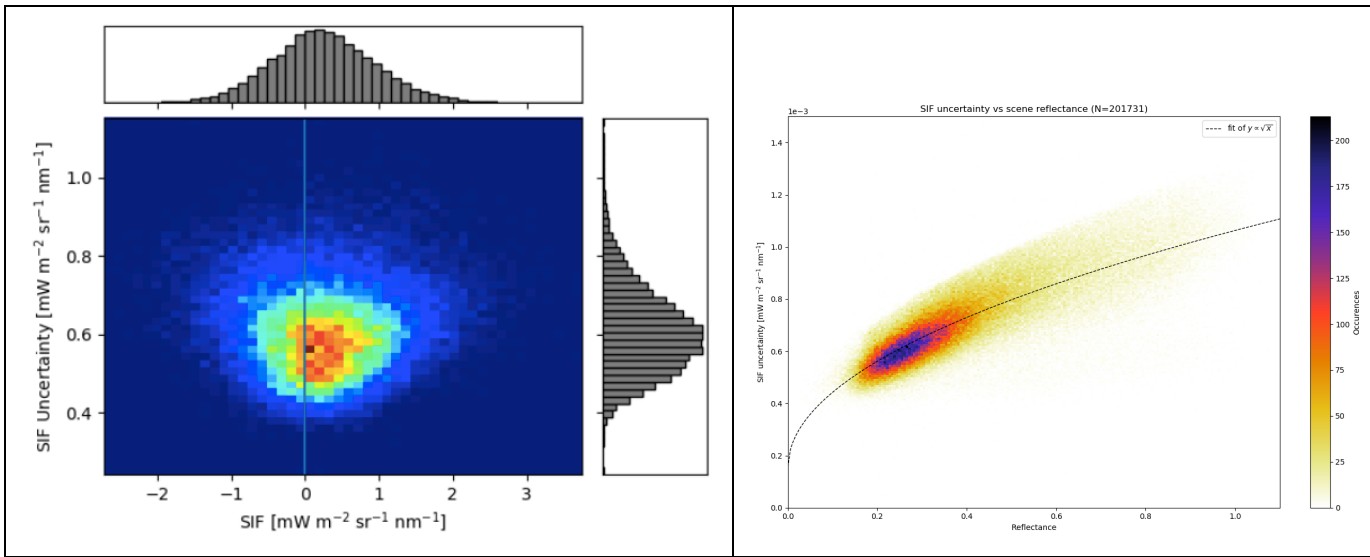

**Figure 7**. Left: uncertainties in GOME-2A fluorescence retrievals as a function of signal strength on 1 August 2012. Only pixels over land (FRESCO+ albedo (at 758 nm) > 0.08) with cloud fractions < 0.4 have been selected. The colours are indicating the number of occurrences of combinations of SIF and SIF uncertainty values. Right: SIF uncertainties as a function of scene reflectance for August 2012.

### 4.6 Data filtering and recommended usage

We recommend using the SIFTER v2 data after applying the autocorrelation filter. Both synthetic and real retrievals are of much better quality when spectral autocorrelation <0.2. This filter rejects faulty retrievals, where too much (structured) spectral residual remains after fitting as discussed in Section 3.2 and in the Section 2 of the Supplemental information. The filter also greatly reduces the standard deviation in the monthly mean retrieved fluorescence, indicating that it successfully removes unrealistically high or low fluorescence values. On average, the autocorrelation filter rejects 5-10% of the retrievals. The majority of the rejected pixels are overlapping with the South Atlantic Anomaly, and to lesser extent with very moist vegetated regions in the tropics (Figure 3.12 in van Schaik [2016]).

The retrievals have been done for scenes with (FRESCO) cloud fractions < 0.4. With such a cloud fraction, there is sufficient light coming from the Earth's surface to effectively detect fluorescence [Frankenberg et al., 2014; Guanter et al., 2015]. Users can expect better quality retrievals for scenes with the lowest cloud fractions, as the surface fluorescence signal then propagates unhindered to the sensor (e.g. Köhler et al., [2015]).

### 5 Retrieval results

### 5.1 Comparison between SIFTER v2 and SIFTER v1

We first examine SIF from SIFTER v2 against our previous SIFTER v1 algorithm [Sanders et al., 2016] for January and July 2011. The spatial distribution of monthly mean SIF agrees reasonably well between SIFTER v1 and v2 (Figure S4) with the highest levels of fluorescence in the vegetated regions of the world and low values over the oceans, deserts, and barren

regions. The spatial correlation between the averaged and gridded SIFTER v2 and v1 data is good: $R^2 = 0.77$ (n=162169) in January and $R^2 = 0.79$ (n=163553) in July.

In the Tropics, SIFTER v2 shows a distinctly different SIF pattern from SIFTER v1 with up to 25% higher fluorescence over the Amazon and Kalimantan in SIFTER v2 (Table S5). We attribute the low tropical fluorescence values to the PCs used in SIFTER v1, which have been derived from a one-year ensemble of reference scenes only, and likely do not capture the full range of water vapour concentrations encapsulated in the 6-year (2007-2012) PC set used in SIFTER v2. Moreover, the SIFTER v1 fit window is much wider, and presumably has stronger sensitivity to interference from water vapour and oxygen absorption, which leads to biases in the fluorescence (Table 2). Another reason for the differences between SIFTER v2 and v1, is the use of the autocorrelation filter in the former. On average, the autocorrelation filter rejects <5% of the retrieved pixels, but the vast majority of these rejections are located within the region of the Southern Atlantic Anomaly. This stricter filtering of faulty retrievals also leads to higher SIFTER v2 fluorescence over the north-western Amazon and lower values over eastern Brazil (Figure 3.12 in van Schaik [2016]).

SIFTER v2 fluorescence shows more plausible values (close to zero) over the northern hemisphere landmasses in January, and over the oceans in January and July. SIFTER v1 gives substantial positive fluorescence in absence of active photosynthesis there. The differences can be understood from the absence of a zero-level adjustment in SIFTER v1. The zero-level adjustment tends to reduce fluorescence at latitudes below 20° N (see Figure 5). We also see substantial difference over the Himalayas and deserts. This is related to the negative fluorescence in SIFTER v2 in those regions.

**5.2 Comparison between SIFTER v2 and NASA v2.8**

We now evaluate the SIFTER v2 fluorescence retrievals against those from the most recent NASA suite (NASA v2.8). The NASA v2.8 retrieval algorithm uses the same 734-758 nm spectral fitting window as SIFTER v2, but the number of principal components (12 for NASA), the reference area (cloudy ocean and Sahara for NASA, Sahara for SIFTER v2), reference sector timespan (daily for NASA, multi-year for SIFTER v2), and use of solar irradiance (daily for NASA, fixed Earth-Sun distance corrected for SIFTER v2) differ substantially between the two approaches. We first carried out a global pixel-to-pixel comparison of fluorescence values in the level-2 products for 15 January and 15 July 2011. Figure 8 shows scatter diagrams and histograms of the pixel-by-pixel differences between the SIFTER v2 and NASA v2.8. We find good correlation between the two retrievals ($R^2 = 0.45$ on 15 January and $R^2 = 0.61$ on 15 July 2011), and indications that SIFTER v2 is some 5-10% lower than NASA v2.8 SIF.

The lower panels of Figure 8 show the probability distribution of the differences in fluorescence values between the SIFTER v2 and NASA v2.8 retrievals. The most frequently occurring differences are close to zero, and SIFTER v2 fluorescence is generally slightly lower than NASA v2.8, by <0.1 mW m$^{-2}$ nm$^{-1}$ sr$^{-1}$. On both days, the fluorescence differences resemble a Gaussian distribution function with a width of 0.4-0.5 mW m$^{-2}$ nm$^{-1}$ sr$^{-1}$. We interpret the width of the distribution as caused by the combined, independent SIFTER v2 ($\sigma_S$) and NASA v2.8 ($\sigma_N$) uncertainties, and if we assume that both retrievals are equally uncertain ($\sigma = \sqrt{\sigma_S^2 + \sigma_N^2}$), we arrive at estimates of $\sigma_S$=0.52 mW m$^{-2}$ nm$^{-1}$ sr$^{-1}$ on 15 January and $\sigma_S$=0.47 mW m$^{-2}$ nm$^{-1}$ sr$^{-1}$ on 15 July 2011. These estimates are consistent with the values discussed in section 4.4 for SIFTER v2 and in Joiner et al. [2013] for NASA retrievals.

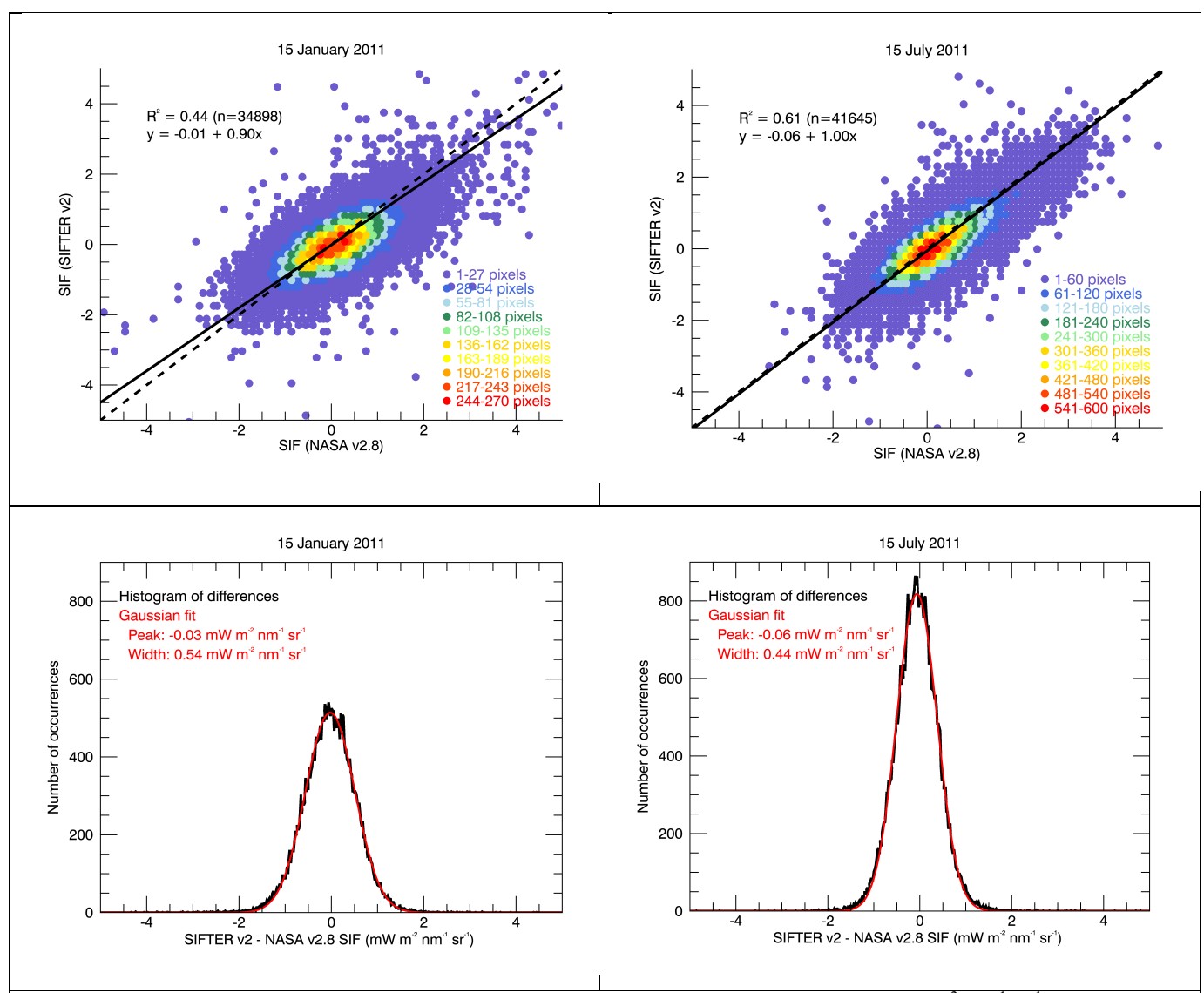

**Figure 8.** Scatterplots of SIFTER v2 vs. NASA v2.8 GOME-2A fluorescence retrievals (in mW m$^{-2}$ nm$^{-1}$ sr$^{-1}$) on 15 January and 15 July 2011. Only collocated pixels over land (albedo > 0.08) with (NASA) cloud fraction < 0.3 have been selected, and pixels were required to be valid in both retrievals. The colours indicate the density of occurrence of combinations of fluorescence values. The legend in the upper panels gives the result of a reduced major axis regression between the fluorescence retrievals.

The spatial distribution of monthly mean SIF agrees well between SIFTER v2 and NASA v2.8 as shown in Figure 9. Both retrievals show very similar patterns with the highest values over South America (Brazil, Bolivia, Argentina) in January, and over the United States corn-belt in July 2011. SIFTER v2 shows slightly negative SIF values over barren and elevated terrain (e.g. western China in January). These negative values can be explained by the choice of reference set as discussed in Botia [2017]. SIFTER v2 has lower fluorescence signals than NASA v2.8 at high latitudes (>60° N) in July, which may also reflect the differences in reference data set selected for the two retrievals.

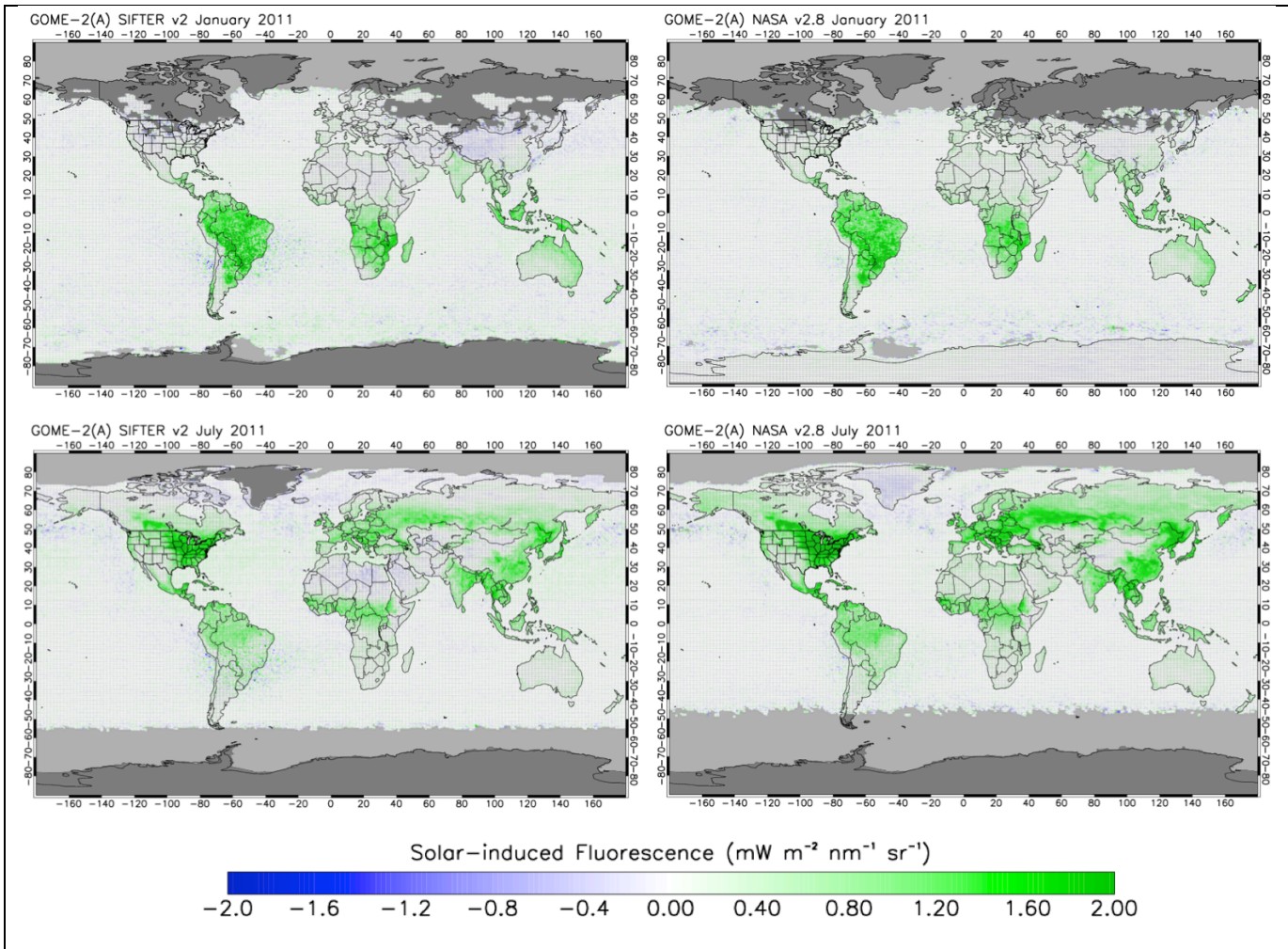

**Figure 9.** Gridded monthly mean SIF values retrieved from GOME-2A with SIFTER v2 (left panels) and NASA v2.8 (right) in January and July 2011. SIFTER data have been selected for autocorrelation < 0.2 and cloud fraction < 0.4, NASA v2.8 data have been selected for cloud fraction <0.3.

We examined the agreement of the monthly mean SIF between our new SIFTER v2 products and the NASA v2.8 product for six vegetated regions across the globe. Within these regions, the two products agree to within 0.4 mW m$^{-2}$ nm$^{-1}$ sr$^{-1}$ and often to within 0.2 mW m$^{-2}$ nm$^{-1}$ sr$^{-1}$. Both data products capture the seasonality of SIF. SIFTER v2 returns 15-35% higher SIF values than NASA v2.8 in the evergreen equatorial regions in January, but 10-30% lower SIF for all six regions in July (Table 5).

**Table 5.** Monthly mean gridded SIF values retrieved from GOME-2A with SIFTER v2 and NASA v2.8 for different vegetated regions throughout the world in January and July 2011.

|  | January 2011 | | | July 2011 | | |
|---|---|---|---|---|---|---|
|  | SIFTER v2 (mW m$^{-2}$ nm$^{-1}$ sr$^{-1}$) | NASA v2.8 (mW m$^{-2}$ nm$^{-1}$ sr$^{-1}$) | Relative difference | SIFTER v2 (mW m$^{-2}$ nm$^{-1}$ sr$^{-1}$) | NASA v2.8 (mW m$^{-2}$ nm$^{-1}$ sr$^{-1}$) | Relative difference |
| Amazon (70°-55°W; 0°-15° S) | 1.47 | 1.27 | +16% | 0.83 | 0.95 | -13% |
| Sub-Saharan Africa (10°W- | 0.21 | 0.31 | N.A. | 0.91 | 1.00 | -9% |

| | | | | | | |
|---|---|---|---|---|---|---|
| 30°E; 5°-10°N) | | | | | | |
| Kalimantan (110°E-115°E; 4°S-6°N) | 0.76 | 0.57 | +34% | 0.49 | 0.58 | -16% |
| United States Cornbelt (96°W-81°W; 38°N-46°N) | -0.09 | 0.07 | N.A. | 1.57 | 1.91 | -18% |
| Western Europe (2°W-15°N; 44°N-52°N) | 0.07 | 0.19 | N.A. | 0.78 | 1.14 | -32% |
| Southeastern China (100°E-120°E; 25°N-35°N) | 0.04 | 0.16 | N.A. | 1.10 | 1.37 | -20% |

**5.3 Impact of the degradation correction on SIF time series**

Figure 10 shows timeseries of SIFTER v2 SIF values retrieved with and without the correction for throughput reductions for the six vegetated regions in Table 5. The corrections are applied on level-1 data retrieved from version 6.0 onwards, i.e. after June 2014. While the figure does not provide direct evidence that the SIFTER v2 data can be used safely for long-term trend analysis, it does suggests that the level-1 degradation correction helps to stabilize our SIF retrievals in later years. A future long-term validation exercise could identify whether spurious trends exist in our SIFTER v2 retrievals.

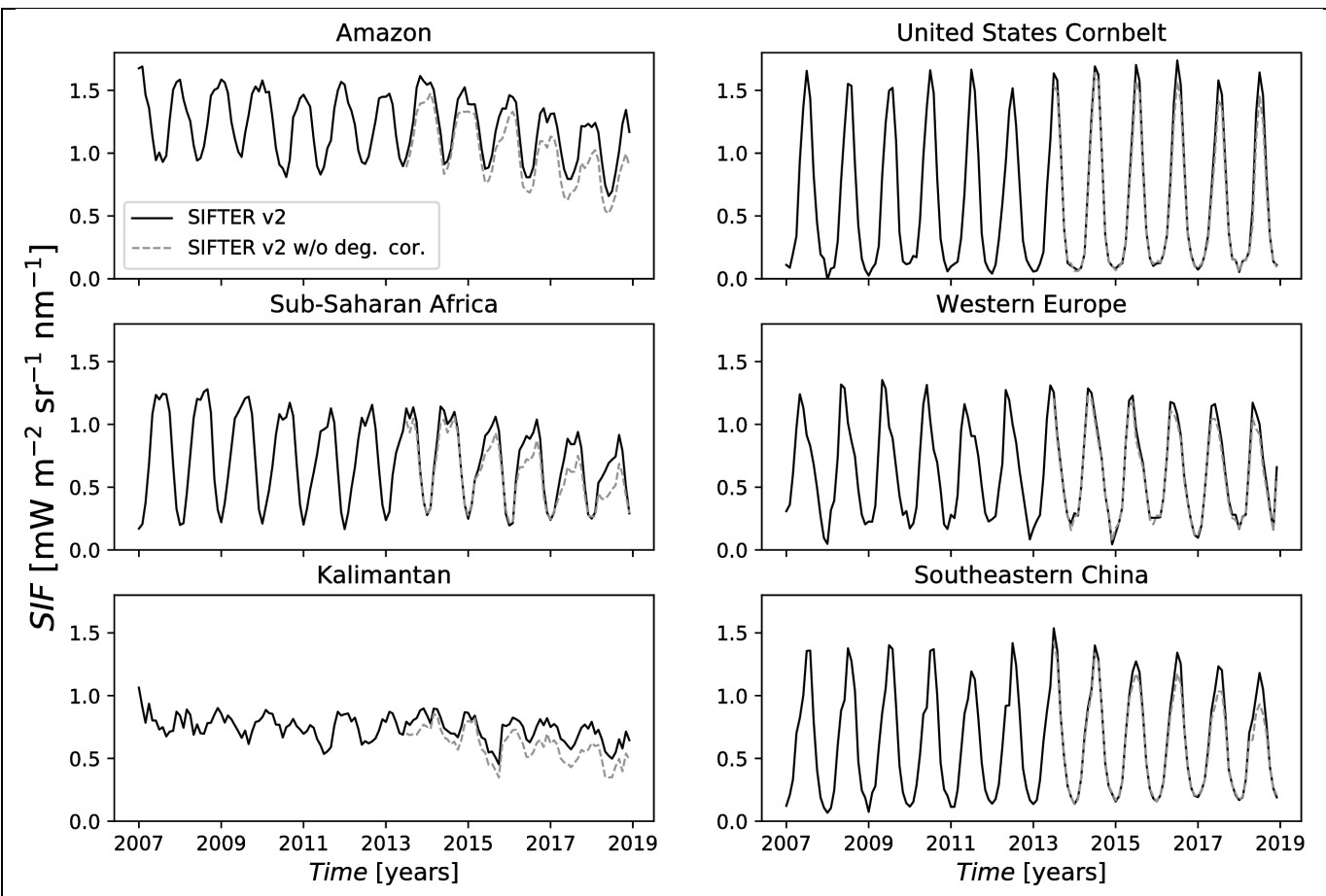

**Figure 10**. Timeseries of GOME-2A SIF retrieved with the SIFTER v2 algorithm with (solid black line) and without degradation correction (grey dashed line) over the 6 vegetated areas defined in Table 5. The degradation correction in our SIFTER v2 dataset are applied only for retrievals based on level-1 data retrieved from version 6.0, 6.1, and 6.2, i.e. from June 2014 onwards.

## 6. Conclusions

We exploited end-to-end tests with the DISAMAR radiative transfer model to inform improvements for our new Sun-Induced Fluorescence of Terrestrial Ecosystems Retrieval version 2 (SIFTER v2) retrieval algorithm. The settings that
optimize retrieval of accurate fluorescence strength are:
- a smaller spectral fitting window (734-758 nm) compared to v1 (712-783 nm). The smaller window excludes the $O_2$-A band and most of the water-vapor absorption features,
- a reduction in the number of principal components (PCs) to describe atmospheric transmission effects to 10 (v2) from 35 (v1). Fewer PCs are needed to explain atmospheric transmittance since $O_2$-A and water vapor features no
longer need to be explained in the fitting procedure,
- the calculation of the PCs reference set from all top-of-atmosphere spectra taken over the non-vegetated parts of the Saharan desert in the entire 2007-2012 period, when the quality of the GOME-2A level-1 data was stable, instead of from the twelve months preceding the GOME-2A measurement of interest, and
- application of a zero-level adjustment by correcting each retrieval within a 0.5° latitudinal bin with the mean
retrieved fluorescence obtained over a Pacific Ocean reference sector within that bin. Such a zero-level adjustment was lacking in SIFTER v1.

The end-to-end tests indicated that the GOME-2A sensitivity to SIF (infilling) is much reduced within the $O_2$-A absorption bands compared to the 740-758 nm, where sensitivity is close to 100%. Spectral analysis of the infilling of the Fraunhofer

lines over a wide spectral window with different sensitivities to SIF therefore becomes problematic. This can be diagnosed by spectral structure in the fit residuals is a strong indicator for unrealistic fluorescence estimates. Such faulty retrievals are effectively filtered out by requiring that spectral autocorrelation in the fitting residuals (or $\chi^2_{red}$) stays below a certain threshold value.

With our improved SIFTER v2 algorithm we generated a multi-year global dataset of solar-induced fluorescence from GOME-2A. The fluorescence data is most reliable when retrieved under clear-sky conditions, when there is sufficient light coming from the Earth's surface, and for low spectral autocorrelation in the fit residuals. We have most confidence in the data quality for the period January 2007 – July 2013, when the reflectance levels were stable and processed with one and the same level-1 processor version (5.3). After the detector switch in July 2013, we applied spectral correction factors to account for the degradation in the level-1 data that is apparent in later processor versions (v6.0 and beyond). Degradation correction has a stabilizing effect on the SIF values retrieved since May 2014 with the introduction of the v6 processor versions. The SIFTER v2 (level-2) data product contains an uncertainty estimate associated with each valid retrieval. The uncertainty for individual retrievals is on the order of 0.6 mW m$^{-2}$ sr$^{-1}$ nm$^{-1}$ with fluorescence signal strengths typically between 0.0-4.0 mW m$^{-2}$ sr$^{-1}$ nm$^{-1}$. The SIFTER v2 dataset shows strongest fluorescence in tropical vegetated regions and in the agriculturally productive regions in northern mid-latitudes. The SIFTER v2 product captures the seasonal variability in step with photosynthetic activity, with near-zero fluorescence in Winter and high fluorescence in Summer over northern mid-latitudes. The limitation of the GOME-2A fluorescence dataset to be used for long-term trend analysis is in the quality of the level-1 data, and the representativeness of the reference spectra used in the retrieval.

We evaluated SIFTER v2 fluorescence against our previous product and against the NASA v2.8 retrievals. Compared to SIFTER v1, SIFTER v2 returns higher fluorescence over strongly vegetated areas, and shows a more pronounced seasonal cycle. The higher tropical fluorescence values in SIFTER v2 are the consequence of using a narrower fit window, a reference set based on a longer period (2007-2012) to capture a more realistic range of water vapour concentrations than in SIFTER v1, and application of a more strict spectral auto-correlation filter. The smaller window implies that SIFTER v2 is less prone to water vapour interference, while the 6-year reference set provides the retrieval with a more complete range of possible water vapour corrections. The more realistic near-zero fluorescence over the oceans and vegetated areas in wintertime are the result of the zero-level adjustment applied in SIFTER v2. NASA v2.8 and SIFTER v2 show a large degree of consistency in capturing spatial and temporal variability in fluorescence. Spatial correlation coefficients between the two level-2 products are high even on the day-to-day level (0.45-0.60). Both retrievals capture maximum fluorescence over the vegetated tropics and agricultural regions in the northern hemisphere in Summer. In the evergreen tropics, SIFTER v2 fluorescence is 15-35% higher than NASA v2.8, but over the summertime mid-latitudes SIFTER v2 is 10-20% lower. Pixel-by-pixel differences between SIFTER v2 and NASA v2.8 retrievals allow evaluation of the combined uncertainties in the retrievals. These correspond to uncertainties of 0.5 mW m$^{-2}$ sr$^{-1}$ nm$^{-1}$ for both retrievals if we assume their uncertainties to be independent and contribute equally. This statistical estimate is slightly lower than the theoretical uncertainty estimate included in the data product.

**Code/Data availability**

SIFTER v2 fluorescence data from GOME-2A is publicly available via [www.temis.nl](www.temis.nl). The SIFTER v2 dataset is linked to a digital object identifier (DOI) [Kooreman et al., 2020]..

**Author contributions**

EvS and FB designed and performed the sensitivity tests, improved the SIFTER v1 retrieval approach, and developed the

zero-level adjustment and auto-correlation filter. MK drafted the algorithm description, performed the theoretical uncertainty analysis, processed the retrieval algorithm to the entire GOME-2A dataset, and analysed the results of retrieval settings. FB analysed the degradation of the GOME-2A level-1 data, analysed the zero-level adjustment, designed the spectral degradation correction, and compared SIFTER v2 to SIFTER v1 and NASA v28 retrievals. FB wrote the manuscript. EvS, MK, and FB contributed equally to this work. PS, OT, and GT supported the processing and analysis of GOME-2A level-1 data at KNMI. AS and WV designed and developed the code and processed the data for SIFTER v1, which was used as the basis for SIFTER v2. RL and AC helped interpret the degradation in GOME-2A level-1 data. JJ and WP provided insight in retrieval sensitivities and suggestions for algorithm improvement. All authors read the manuscript and provided feedback that improved the paper.

**Competing interests**

The authors declare no competing interests.

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
