# Peer review of "Improved SIFTER v2 algorithm for long-term GOME-2A satellite retrievals of fluorescence with a correction for instrument degradation"

_Atmospheric Measurement Techniques, 2019_

## Referee Comment (RC1) · Anonymous Referee #1 · 6 Feb 2020

Genereral comment

The manuscript entitled with "Improved SIFTER v2 algorithm for long-term GOME-2A satellite retrievals of fluorescence with a correction for instrument degradation" by van Schaik et al. tackles some interesting technical aspects related to the GOME-2 SIF retrievals. In particular, the assessment of degradation issues is crucial for the interpretation of the resulting data set. Improvements were apparently achieved by narrowing the fitting window, resulting in a very similar retrieval set-up with respect to the NASA algorithm. Therefore, it is not surprising that the results converge, but reassuring. While the manuscript is generally well written, there are some important

corrections necessary and I recommend to add some additional analyses. My major concern relates to the end-to-end test, where the retrieval seems to perform worse than for the real data, even though the simulations are representing the ideal case (noise-free). To identify any fitting issues (over or underfitting), adding a reduced Chi square anaylsis (expected vs observed) would be necessary, at least for the real data.

Specific comments

P2L9-10 van der Tol (2014) claims that SIF is driven by $CO_2$ concentration? This is certainly wrong, the $CO_2$ concentration is relevant for the dark reaction of photosynthesis.

P2L22-24 overly complicated sentence. Please rephrase

P4L20 'understood' → is due to

P6L33 – P7L3 What justifies this particular interpretation? What means "and so on"? A physical meaning may be attached to the PCs, but this is not always the case. Here, all PCs show features related to oxygen and water vapour. Maybe remove this sentence?

P7L3 This sentence has a trivial meaning, masking the important details of how many components are actually required to model the transmission with sufficient accuracy and if all of the PCs are indeed needed. This is relevant, because it is known that the number of PCs has effects on the retrieval accuracy and precision, reported by Guanter et al. (2013), Joiner et al. (2013), and Köhler et al. (2015).

P7Eq2 The forward model is written in a way that assumes SIF as known. If, as claimed, there is one fitting parameter for SIF, there should be a normalized spectral shape in combination with one coefficient to fit.

P7L15 The spectral shape of SIF in Daumard et al. (2012) is based on leaf level measurements, not "spectral measurements over various vegetated fields". "...considerable uncertainty on the shape..." is inaccurate. In fact, Magney et al. (2019) showed that the spectral shape of SIF in the far-red is remarkably stable across species and environmental conditions. However, in wavelengths < ∼730nm the shape is controlled by re-(absorption) effects, which is the origin of the uncertainty.

P7L17 The sentence about the spectral shape is misleading as the two tested spectral shapes in Parazoo et al. (2019) are very similar. The main message here should be that the fitting window ideally covers the spectral region where SIF is stable, otherwise re-absorption effects may interfere with the retrieval. However, this limits the number of spectral points, which increases the retrieval noise. Conversely, a wider retrieval window makes the retrieval less noisy, but affects the retrieved SIF magnitude (introduces a bias). The overlap with solar Fraunhofer lines does not change the retrieved magnitude, it is a necessary prerequisite. Please rephrase according to my comment.

P7L21 This sentence appears to be out of context.

P7L28 I don't understand this sentence. What means "...so orthogonality between the albedo plus two-way transmission and SIF plus one-way transmission terms is important."?

P7L30 I don't think that an "Internship Report" is a legit, citeable reference. Unrealistic albedo or transmittance values occur under "some viewing conditions" and lead to negative SIF estimates? This raises several questions and appears to be out of context, please remove this sentence entirely.

P8L20 The opening of the sentence sounds odd. Replace "our understanding" by "our assumptions to model ..."? Provide a reference for the DISAMAR radiative transfer model, the abbreviation has not been introduced.

P9Fig.2 Please check your y-axis, spelling and unit. What does absorption > 1 mean? Explained variances could be added.

P10L9 This sentence is odd. First, "spectra" is misleading, because PCs are compared. Second, it should be self-explanatory that a radiative transfer model is able to "capture the various relevant atmospheric processes".

P10L11 The informative value of comparing PCs from observations and simulations is not obvious to me. The sign could be flipped, higher order PCs explain only a fraction of the variance and might occur at different positions (e.g. PC#3 from simulations could be PC#5 in observations). Would you clarify what this comparison should tell the reader and why it deserves a Figure? Furthermore, why should we expect the PCs to be similar if the real measurements include noise while the simulations seem to be noise-free?

P10L20 As there is no word about noise in the simulations, I have to assume that they are noise free, in which case the poor retrieval performance shown in Table 2 is surprisingly disappointing. Without noise, the RMSE could be zero.

P12-13 Even though it is interesting to explore alternative ways to identify poor retrievals, the authors should consider to use the reduced Chi square instead. The reduced Chi square would reveal any deficiencies in the retrieval set-up.

P13L9-21 This paragraph reveals that there is a serious issue when modeling water vapour absorption. This is an idealized scenario in which 65% of the retrievals are flagged as poor. From my perspective, this is far from being acceptable.

P14L21 Is there a reference to justify that the Saharan desert has a "dynamic range" of water vapour in the first place?

P15L5 "but not dramatically so"? This seems to be a bold claim.

P17L15 Could you elaborate on the exact mechanism by which the slit function introduces a bias or is this hypothesis purely based on the latitudinal offset?

P18L5 Could you add a Figure to illustrate the bias correction? I am particularly interested to see if a linear fit with radiance levels is justified.

P18L10 Does this sentence mean that you apply a daily and monthly bias correction? Please clarify.

P18L22-26 Again, it is not appropriate to cite an "Internship Report" which is not publicly available.

P20L1 Could you explain how the degradation relates to the different L1 processors? Please discuss what kind of effect could potentially result from the change in L1 processors.

P21L11 Again, "Internship Report"

P21L14 Is there a reason to assume that the uncertainty is driven by the SIF signal level? It should be driven by the radiance level. Please add (or replace) the uncertainty vs radiance level in Figure 6.

P22L6 Replace "yields" by values. How come that the filter rejects suddenly only 5-10% of the measurements? This is better than in all of the experiments.

P27Fig.9 The Amazon time series still shows a significant SIF decrease after 2013. Even though the reliability for the later period is already extensively discussed, are there other potential sources for artificial trends?

P27L6 I see this as an overstatement. No retrieval algorithm has been developed, but different parameters have been optimized.

References:

Guanter, L., Rossini, M., Colombo, R., Meroni, M., Frankenberg, C., Lee, J.-E., & Joiner, J. (2013). Using field spectroscopy to assess the potential of statistical approaches for the retrieval of sun-induced chlorophyll fluorescence from ground and space. Remote Sensing of Environment, 133 , 52–61.

Joiner, J., Guanter, L., Lindstrot, R., Voigt, M., Vasilkov, A., Middleton, E., Frankenberg, C. (2013). Global monitoring of terrestrial chlorophyll fluorescence from moderate-spectral-resolution near-infrared satellite measurements: methodology, simulations, and application to GOME-2. Atmospheric Measurement Techniques, 6 (10), 2803–

2823.

Köhler, P., Guanter, L., & Joiner, J. (2015). A linear method for the retrieval of sun-induced chlorophyll fluorescence from GOME-2 and SCIAMACHY data. Atmospheric Measurement Techniques, 8 (6), 2589–2608.

Magney, T. S., Frankenberg, C., Köhler, P., North, G., Davis, T. S., Dold, C. (2019). Disentangling changes in the spectral shape of chlorophyll fluorescence: Implications for remote sensing of photosynthesis. Journal of Geophysical Research: Biogeosciences.
* * *

---

## Referee Comment (RC2) · Anonymous Referee #2 · 17 Apr 2020

1. What is the main achievement obtained through this study (it sounds more like a technical report)? 2. "Our results support the use of SIFTER v2 data to be used as an independent constraint on photosynthetic activity on regional to global scales." Where is this justified from the current paper?

While this study provides a well written overview of the improved algorithm for SIF retrievals, I am worried that it lacks original scientific content. The majority of the work presented is incremental in nature and reads, to the most part, like a well-written tech report. What I am missing are real scientific discussions on WHY these algorithm changes are important, why the omission of some absorption bands is so crucial (see

later, this is a hot topic, it would be good to discuss this), etc. I just feel that the authors need to do a better job in outlining what is really part of their original work and what is not. Adding a more in-depth case study as to why the O2 band is hurting the retrieval might help fill the gap in current originality. I don't want to be overly demanding but as nice as the paper reads at the moment, the authors have to be honest and clearly outline what is original vs. a reproduction of most basic concepts of the Joiner and Koehler approaches. Without this, it would remain a tech report only and I would have to defer the publication decision to the editor.

Please find a few more detailed comments below:

SIFTER needs to be explained in abstract already.

P2 Line 4: 19% released as heat: This sounds oddly specific. In fact, the heat quenching is highly variable, please make this clear

P2 Line 9: SIF doesn't know about the CO2 concentration (at least not directly). See Rev #1.

P2 Line 31: KNMI needs to be explained in the first instance (not everyone knows it).

P6-7: Lines 23++: Are any of these steps new? Which ones are identical to Joiner et al? Which ones are identical to SIFTER 1? Which ones come from this paper?

P7 -Lines 21-22: Maybe a copy&paste error? What does that sentence mean anyhow, I am not really sure (too vague).

P7 L30: Please make clear that the impact of transmission function is only of importance for the large spectral windows as needed for GOME-2

P8-Lines 6++: If I look at the list of changes made here, they really look rather incremental, fine-tuning some retrieval settings. The big question is whether this warrants publication in a peer reviewed journal as "original" work. You will have to justify this to some degree. (i.e. why is this more than an internal tech report?)

P8-Line 12: I think it is important to underline (again) that the exclusion of the O2-A band actually helps the retrieval. This is the topics of a long-standing debate and the original algorithm included the O2A band under the assumption that it "helps" the retrieval. It would be good to elaborate more on that specific issue and show the community clearly why it harms the actual retrieval. This could be a valuable addition.

P13-Lines 25 and around: Again, you are converging to a similar fitting window as Joiner et al did for a long time already. The only thing new is that you provide some more tests (of which I am sure Joiner did as well).

Page 15, lines 4++ Temperature can have a profound effect on water vapor absorptions (due to a wide range of lower state energies), just using the Sahara might under-represent this change in spectroscopy. P18 L 1++: Why should the bias depend on latitude at all (and not, say Air Mass Factor alone)? 1 degree also sounds really fine. What I am missing in most discussions is the lack of a mechanistic motivation for certain choices. Why is the latitude dependent bias not symmetric? Why do you assume it is there in the first place?

P21,Figure 6: I don't understand what this figure is telling me. Why "should" the uncertainty depend on the SIF signal? What is being tested here? I am a bit lost. What would make sense is to plot the uncertainty against continuum level radiance, thus to the overall SNR of the spectrum. Absolute SIF will get noisier at higher signals (which can then lead to some correlations of SIF uncertainty and SIF signal as vegetation is very bright in the NIR). Here, however, the single pixel sigma is used, which is so large that it is hard to see these effects.

Section 5.2: Wo what does this tell us? It is all purely descriptive. Sentences like "We find that both data products capture the seasonality of SIF, which suggests that actual fluorescence in response to photosynthesis is being measured" are rather vague. Is the final key point of the paper that you managed to reproduce the Joiner retrievals?

---

## Author Comment (AC1) · 15 Jun 2020

We thank the reviewer for this assessment and for his/her overall insightful and constructive review that has helped us to improve the manuscript. The reviewer's comment is in normal black font, our response is in blue font. In the accompanying revised manuscript the applied corrections are visible in the track changes.

The manuscript entitled with "Improved SIFTER v2 algorithm for long-term GOME-2A satellite retrievals of fluorescence with a correction for instrument degradation" by van Schaik et al. tackles some interesting technical aspects related to the GOME-2 SIF retrievals. In particular, the assessment of degradation issues is crucial for the

interpretation of the resulting data set.

Improvements were apparently achieved by narrowing the fitting window, resulting in a very similar retrieval set-up with respect to the NASA algorithm. Therefore it is not surprising that the results converge, but reassuring. While the manuscript is generally well written, there are some important corrections necessary, and I recommend to add some additional analyses. My major concern relates to the end-to-end test, where the retrieval seems to perform worse than for the real data, even though the simulations are representing the ideal case (noise free). To identify any fitting issues (over or underfitting), adding a Chi square analysis (expected vs observed) would be necessary, at least for the real data.

The simulations in the end-to-end test included noise, and represented as much as possible scenarios that are also encountered in reality, so with noise and high water vapour concentrations that exceed the range represented in the principal component spectra. We have now added a Chi square analysis in the discussions and also in the retrieval product.

Specific comments P2L9-10 van der Tol (2014) claims that SIF is driven by $CO_2$ concentration? This is certainly wrong, the $CO_2$ concentration is relevant for the dark reaction of photosynthesis.

Point taken. We removed this part of the sentence. The photochemical yield is influenced by the rate of $CO_2$ fixation by plants, and this rate is not directly related to the ambient $CO_2$ concentration.

P2L22-24 overly complicated sentence. Please rephrase

We rephrased the sentence.

P4L20 'understood' → is due to

Agreed. We changed to 'is due to'.

P6L33-P7L3 What justifies this particular interpretation? What means "and so on"? A physical meaning may be attached to the PCs, but this is not always the case. Here, all PCs show features related to oxygen and water vapour. Maybe remove this sentence?

We compared principal component spectra collected over the non-vegetated Sahara, and noticed that the first 4 PCs were very similar between GOME-2 and DISAMAR simulations. PC1 represents the mean, and PC2 mostly represents the variability caused by water vapour. We agree that a physical meaning is difficult to assign to PCs 3 and further, so we removed the "and so on", as indeed the PCs show features related to variability caused by $H_2O$, $O_2$, and also other issues (noise, unresolved spectral residuals). We rephrased accordingly.

P7L3 This sentence has a trivial meaning, masking the important details of how many components are actually required to model the transmission with sufficient accuracy and if all of the PCs are indeed needed. This is relevant, because it is known that the number of PCs has effects on the retrieval accuracy and precision, as reported by Guanter et al. (2013), Joiner et al. (2013), and Kohler et al. (2015).

Our tests with real GOME-2 data also showed that the choice for the number of PCs is important, with 10 being the optimal choice (most robust patterns, highest precision) for our retrieval configuration with a constant number of PCs for each retrieval. We now include: *"Higher order PCs represent variability caused by water vapour, oxygen, and from other sources such as noise, unresolved surface and instrumental effects."*

P7Eq2 The forward model is written in a way that assumes SIF as known. If, as claimed, there is one fitting parameter for SIF, there should be a normalized spectral shape in combination with one coefficient to fit.

Thanks for spotting this. There should indeed be a fitting coefficient term in Eq. (2) to reflect that the fit model allows the SIF-contribution to vary for different scenes. We updated Eq. (2) by introducing the fit parameter *c* in Eq. 2.

P7L15 The spectral shape of SIF in Daumard et al. (2012) is based on leaf level measurements, not "spectral measurements over various vegetated fields". "...considerable uncertainty on the shape..." is inaccurate. If fact, Magney et al. (2019) showed that the spectral shape of SIF in the far-red is remarkably stable across species and environmental conditions. However, in wavelengths $< \sim 730$ nm the shape is controlled by re-(absorption) effects, which is the origin of the uncertainty.

Daumard et al. [2012] measured fluorescence fluxes at two different wavelengths: 685 nm and 760 nm, and obtained fluorescence spectra based on these measurements. Their fluorescence spectra show variability especially below 730 nm as driven by chlorophyll concentration, a conclusion in line with Magney et al. [2019]. We updated the text accordingly.

P7L17 The sentence about the spectral shape is misleading as the two tested spectral shapes in Parazoo et al. (2019) are very similar. The main message here should be that the fitting window ideally covers the spectral region where SIF is stable, otherwise re-absorption effects may interfere with the retrieval. However, this limits the number of spectral points, which increases the retrieval noise. Conversely, a wider retrieval window makes the retrieval less noisy, but affects the retrieved SIF magnitude (introduces a bias). The overlap with solar Fraunhofer lines does not change the retrieved magnitude, it is a necessary prerequisite. Please rephrase according to my comment.

Agreed. We now adapted the text to read: *"A careful trade-off is required in the selection of the fitting window: it should preferably overlap with the stable part of the fluorescence reference spectrum (less uncertainty), contain several Fraunhofer lines, but preferably avoid strong absorption features from oxygen and water vapour, as we will see."*

P7L21 This sentence appears out of context.

We removed the sentence.

P7L28 I don't understand this sentence. What means "...so orthogonality between the albedo plus two-way transmission and SIF plus one-way transmission terms is important."?

We removed this sentence.

P7L30 I don't think that an "Internship Report" is a legit, citeable reference. Unrealistic albedo or transmittance values occur under "some viewing conditions" and lead to negative SIF estimates? This raises several questions and appears to be out of context, please remove this sentence entirely.

Since the research described in that internship report (an open source, but grey literature) does not help the clarity of the paper here, we have now removed these sentences.

P8L20 The opening of the sentence sounds odd. Replace "our understanding" by "our assumptions to model..."? Provide a reference for the DISAMAR radiative transfer model, the abbreviation has not been introduced.

Thanks for the suggestion: "assumptions to model" indeed captures better what we intend to say. A reference for DISAMAR, De Haan [2011], has now been added, and the acronym has been written out.

P9Fig. 2 Please check your y-axis, spelling and unit. What does absorption > 1 mean? Explained variances could be added.

We corrected the spelling of y-axis in Figure 2. Absorption refers to the two-way slant optical thickness caused by the absorption of light by water vapour and oxygen. We adapted the caption of Figure 2 accordingly.

P10L9 This sentence is odd. First, "spectra" is misleading, because PCs are compared. Second, it should be self-explanatory that a radiative transfer model is able to "capture the various relevant atmospheric processes".

The sentence refers to the Figure 2 that compares slant atmospheric absorption (optical thickness) spectra, so we don't think that that is misleading. As concerns the second point, we adapted the sentence and removed the part that DISAMAR captures the various relevant processes.

P10L11 The informative value of comparing PCs from observations and simulations is not obvious to me. The sign could be flipped, higher order PCs explain only a fraction of the variance and might occur at different positions (e.g. PC3 from simulations could be PC5 in observations). Would you clarify what this comparison should tell the reader and why it deserves a Figure?

The comparison was carried out to obtain confidence that we understand the main drivers of variability in the reference spectra, and to confirm that we can use DISAMAR as a tool to test retrieval settings. This was not obvious at first, but we agree that the (lower panel of) Figure 2 and associated text are better moved to the Supplement, which is what we have now done.

P10L20 As there is no word about noise in the simulations, I have to assume that they are noise free, in which case the poor retrieval performance shown in Table 2 is surprisingly disappointing. Without noise, the RMSE could be zero.

The caption of Table 1 mentions that "signal-to-noise ratio of the simulated reflectance spectra was 1,000". We now also included that information in the caption of Table 2.

P12-13 Even though it is interesting to explore alternative ways to identify poor retrievals. The authors should consider to use the reduced Chi square instead. The reduced Chi square would reveal any deficiencies in the retrieval set-up.

We thank the reviewer for the suggestion. Besides the metrics included in Table 2, we checked the reduced Chi squared metric to ensure that the retrieval is not suffering from overfitting or underfitting. We found that the non-faulty retrievals all have values close to 1, and faulty retrievals generally have a $\chi^2_{red}$ >3. We now included this information in

the main text, and report $\chi^2_{red}$ in the SIFTER v2 data product.

P13L9-21 The paragraph reveals that there is a serious issue when modelling water vapour absorption. This is an idealized scenario in which 65

As indicated in Table S2, the scenario is not exactly 'ideal'. Our purpose was to test the robustness of the fitting windows for situations with a much moister atmosphere (30-65 g m$^{-2}$) than in the reference PC set (4-40 g m$^{-2}$). The faulty cases occur for those scenes where water vapour > 40 g m$^{-2}$. For non-faulty retrievals, the 734-758 nm window provides the most unbiased results. In an additional test, we performed the retrieval with a PC set based on 30-65 g m$^{-2}$ water vapour columns, and the number of faulty retrievals reduced strongly. We now explain this in the manuscript.

P14L21 Is there a reference to justify that the Saharan desert has a "dynamic range" of water vapour in the first place?

We found the dynamic range in the ECMWF ERA-Interim meteorological fields [Dee et al., 2011], which have been evaluated for water vapour with satellite measurements in Grossi et al. [2015]. These references have been included in the manuscript.

P15L5 "but not dramatically so"? This seems to be a bold claim.

The ECMWF data suggest that by column, water vapour over the Sahara is between 20-80% lower than over tropical forests. We removed the "dramatically so".

P17L15 Could you elaborate on the exact mechanism by which the slit function introduces a bias or is this hypothesis purely based on the latitudinal offset?

The GOME-2A slit function is known to change significantly over time because of temperature changes [Munro et al., 2016]. Changes in the shape of the slit function are known to have caused highly structured spectral responses and thereby problems with the fitting of HCHO, $O_3$, and $NO_2$ in the UV-Vis part of the GOME-2A spectra (e.g. De Smedt et al. [2012], Miles et al. [2015], Azam and Richter [2015], Beirle et al. 2017]). These changes occur along an orbit (thus with latitude) and (we now include

*in the manuscript text) "appear as an increase in the width of the slit function, with implications for the depth of the Fraunhofer structures: the wider the slit, the less deep the Fraunhofer lines. Shallower Fraunhofer lines may then be interpreted by the fitting algorithm to have been caused by fluorescence, which explains the positive fluorescence bias in the southern hemisphere (Figure 4). Conversely, sharper and deeper Fraunhofer lines (relative to the width of the lines over the Sahara), may well be interpreted as caused by negative fluorescence, explaining the negative bias for latitudes north of the Sahara."*

The problems have been mitigated in the trace gas algorithms by extending the fitting approach with dynamical fit parameters describing the width and shape of the slit function, with good results [Beirle et al., 2017]. Such an approach could also be attempted for GOME-2 SIF retrievals in the far-red part of the spectrum. These are limited by the reference spectra taken over the middle of the orbit, when the slit function has an intermediate width that is likely not representative for smaller widths north of 30°N, and the larger widths in the southern hemisphere.

P18L5 Could you add a Figure to illustrate the bias correction? I am particularly interested to see if a linear fit with radiance levels is justified.

The fit of the SIF zero level offset against radiance level is mostly a smoothing operation. The new figure S2 (right panel) shows the linear fit of SIF against radiances of all pixels with cloud fractions < 0.4 that have been observed at 45°N between 130-150 °W in July 2007. There is a weak relationship that suggests a more prominent non-zero bias for high radiance levels, i.e. when Fraunhofer lines are relatively well-defined. The mean bias at 45°N based on all pixels is -0.095 mW m$^{-2}$ sr$^{-1}$ nm$^{-1}$ and the median value is -0.088 mW m$^{-2}$ sr$^{-1}$ nm$^{-1}$. For comparison, we also plotted the binned mean SIF values with intervals of $0.1 \times 10^{13}$ photons cm$^{-2}$ s$^{-1}$ sr$^{-1}$ nm$^{-1}$ (light green squares). The linear regression through all data (black circles) follows the binned values quite well, suggesting that a linear fit is reasonable. We now include this figure (Figure 1 below and as Figure S2 in the revised Supplemental Material) and discussion

in the Supplemental Material.

P18L10 Does this sentence mean that you apply a daily and monthly bias correction? Please clarify.

We now include in the manuscript that *"In the operational retrieval, the bias correction is determined based on the most recent days as soon as a sufficient number of pixels has been collected in a latitude bin (at least 10 per bin). In practice, the bias correction is often based on the last day, but for some latitude bands, the correction can be based on pixels dating back at most 14 days."*

P18L22-26 Again, it is not appropriate to cite an "Internship Report" which is not publicly available.

We checked the guidelines of Copernicus Publications and note that "Informal or so-called "grey" literature may only be referred to if there is no alternative from the formal literature. Works cited in a manuscript should be accepted for publication or published already." In this case, we think it is relevant to refer to the Report by van Schaik [2016] which is publicly available via the url provided in the reference list.

P20L1 Could you explain how the degradation relates to the different L1 processors? Please discuss what kind of effect could potentially result from the change in L1 processors.

Figure 5(b) and the consistent trends in reflectance over Libya4 (Figure 1) suggest that the degradation is mostly progressing in time rather than with level-1 processor version (Table S1). For the changes in processor version, some impact of the change from 6.0 to 6.1 (in June 2015) might be expected – the other 2 changes (from 5.3 to 6.0 and from 6.1 to 6.2) did not affect calibration. For the processing of v6.1 data, the in-flight derived BSDF for solar radiometric calibration was introduced, which may have resulted in changes of the radiometric accuracy to unknown extent. However, we do not find any clear evidence for this in Figures 1, 6, or 10.

P21L11 Again, "Internship Report".

See above.

P21L14 Is there a reason to assume that the uncertainty is driven by the SIF signal level? It should be driven by the radiance level. Please add (or replace) the uncertainty vs radiance level in Figure 6.

Figure 6 shows that the uncertainty is not driven by the SIF signal itself, which is reassuring. This was already discussed in the text. Thanks for the suggestion to also show the uncertainty vs. reflectance level. We now do that in the new Figure 6(b).

P22L6 Replace "yields" by values. How come that the filter rejects suddenly only 5-10

Done. The filter itself is the same. However, the retrieval scenes and reference PC set are different between GOME-2A and the DISAMAR tests. For example, the GOME-2A set contains a much larger set of surface albedo values, viewing geometries, SIF values, etc. A similar rejection rate should therefore not be expected.

P27Fig.9 The Amazon time series still shows a significant SIF decrease after 2013. Even though the reliability for the later period is already extensively discussed, are there other potential sources for artificial trends?

We have recently compared trends in SIF over the Amazon from our GOME-2A SIFTER v2 product to trends in OCO-2 SIF and NIRv from MODIS. The SIFTER v2 product is not showing a decreasing trend relative to these other data products over the Amazon (preliminary results). Other possible errors could be the representativeness of the PC set for later years, albedo trends, and insufficient correction for water vapour absorption.

P27L6 I see this as an overstatement. No retrieval algorithm has been developed, but different parameters have been optimized.

We removed this sentence.

[Figure]

**References**

Azam, F. and Richter, A.: GOME2 on MetOp: Follow-on analysis of GOME2 in orbit degradation, Final report, EUM/CO/09/4600000696/RM,2015, available at: http://www.doas-bremen.de/reports/gome2_degradation_follow_up_final_report.pdf (last access: 7 September 2016), 2015.

Beirle, S., Lampel, J., Lerot, C., Sihler, H., and Wagner, T.: Parameterizing the instrumental spectral response function and its changes by a super-Gaussian and its derivatives, Atmos. Meas. Tech., 10, 581–598, https://doi.org/10.5194/amt-10-581-2017, 2017.

De Smedt, I., Van Roozendael, M., Stavrakou, T., Müller, J.-F.,Lerot, C., Theys, N., Valks, P., Hao, N., and van der A, R.: Im-proved retrieval of global tropospheric formaldehyde columnsfrom GOME-2/MetOp-A addressing noise reduction and instrumental degradation issues, Atmos. Meas. Tech., 5, 2933–2949,doi:10.5194/amt-5-2933-2012, 2012.

Miles, G. M., Siddans, R., Kerridge, B. J., Latter, B. G., andRichards, N. A. D.: Tropospheric ozone and ozone profiles re-trieved from GOME-2 and their validation, Atmos. Meas. Tech.,8, 385–398, doi:10.5194/amt-8-385-2015, 2015.

Munro, R., Lang, R., Klaes, D., Poli, G., Retscher, C., Lind-strot, R., Huckle, R., Lacan, A., Grzegorski, M., Holdak, A.,Kokhanovsky, A., Livschitz, J., and Eisinger, M.: The GOME-2 instrument on the Metop series of satellites: instrument de-sign, calibration, and level 1 data processing – an overview, At-mos. Meas. Tech., 9, 1279–1301, doi:10.5194/amt-9-1279-2016, 2016.

[Figure]

Pacific (130°-150° W)

R = -0.13

$y = 0.12 - 8.3 \ 10^{15} \ x$

SIF (mW m$^{-2}$ sr$^{-1}$ nm$^{-1}$)

Radiance (ph cm$^{-2}$ s$^{-1}$ sr$^{-1}$ nm$^{-1}$)

**Fig. 1.** SIF over the Pacific reference sector against radiances of all pixels with cloud fractions
< 0.4 that have been observed at 45°N in July 2007.

---

## Author Comment (AC2) · 15 Jun 2020

We thank the reviewer for this assessment and for his/her overall insightful and constructive review that has helped us to improve the manuscript. The reviewer's comment is in normal black font, our response is in blue font. In the accompanying revised manuscript the applied corrections are visible in the track changes.

1. What is the main achievement obtained through this study (it sounds more like a technical report)? 2. "Our results support the use of SIFTER v2 data to be used as an independent constraint on photosynthetic activity on regional to global scales". Where is this justified from the current paper?

1. The achievements of our work have been explained in the Abstract: we show a tangible improvement of the previous SIFTER (v1) algorithm presented in Sanders et al. [2016]. As a reminder, the improvements are (a) a spectral fitting approach that reproduces fluorescence in an end-to-end test (more about that below), (b) application of a principle component set based on 5-years worth of reference spectra, and (c) first-ever assessment of GOME-2A reflectance degradation, and a first attempt to correct for this.

2. The scope of this paper is not the use of our SIFTER v2 data to constrain patterns of photosynthetic activity, but rather to show the algorithm mechanics, necessary corrections for spatiotemporal biases, and to evaluate the product and its uncertainty against the NASA SIF retrievals. The fact that a pre-release of SIFTER v2 has already been used to constrain photosynthesis in the Amazon (Koren et al., 2018), and that this pre-release has now improved further (via the degradation correction) is in our view encouraging to users looking for new global proxies for carbon uptake.

While this study provides a well written overview of the improved algorithm for SIF retrievals, I am worried that it lacks original scientific content. The majority of the work is incremental in nature and reads, to the most part, like a well-written tech report. What I am missing are real scientific discussions on WHY these algorithm changes are important, why the omission of some absorption bands is so crucial (see later, this is a hot topic, it would be good to discuss this), etc. I just feel the authors need to a better job in outlining what is really part of their original work and what is not. Adding a more in-depth case study as to why the $O_2$ band is hurting the retrieval might help fill the gap in current originality. I don't want to be overly demanding but as nice as the paper reads at the moment, the authors have to be honest and clearly outline what is original vs. a reproduction of the most basic concepts of the Joiner and Koehler approaches. Without this, it would remain a tech report only and I would have to defer the publication decision to the editor.

We appreciate the reviewer's critical perspective, which moved us to analyse in more

depth why excluding the $O_2$-A band from the spectral analysis for space-based approaches leads to a much better reproduction of the SIF signal in an end-to-end test. We believe that our postulated SIF 'air mass factor' is instructive in understanding how absorption of the SIF signal by $O_2$ between the Earth and GOME-2 distorts the relative depth of the Fraunhofer lines within the absorption band compared to the relative depth of the lines in the full transmission part of the spectral window ($\sim$740-758 nm). This opens up new avenues to further optimize the spectral fitting window in future studies. We now also discuss in more detail why we think the latitude-dependent bias is negative in the northern hemisphere, and positive in the southern hemisphere, also suggesting next steps, such as dynamically fitting the GOME-2A slit function in the non-linear least squares regression.

Please find a few more detailed comments below.

SIFTER needs to be explained in abstract already.

We now explain the abbreviation SIFTER in the abstract.

P2 Line 4: 19% released as heat: This sounds oddly specific. In fact the heat quenching is highly variable, make this clear.

We have now modified the sentence without making reference to seemingly exact percentes: *"Most of the solar energy that a plant receives is used for photosynthesis, but part is released as heat and between 1-2% is re-emitted as fluorescence at longer wavelengths [Baker and Oxborough, 2004]."*

P2 Line 9: SIF doesn't know about the $CO_2$ concentration (at least not directly). See Rev1.

Point taken. We removed this part of the sentence. The photochemical yield is influenced by the rate of $CO_2$ fixation by plants, and this rate is not directly related to the ambient $CO_2$ concentration.

P2 Line 31: KNMI needs to be explained in the first instance (not everyone knows it).

We have now added that the abbreviation refers to Royal Netherlands Meteorological Institute.

P6-7: Lines 23++: Are any of these steps new? Which ones are identical to Joiner et al.? Which ones are identical to SIFTER 1? Which ones come from this paper?

Section 2.2 provides the basic information for the SIFTER algorithm, and in Section 2.3 we have now included more motivation and highlight the delta's between the SIFTER v1 and the NASA algorithm.

The SIFTER v2 algorithm has grown out of two years of collaboration (and exchange of ideas via email and on international meetings) between KNMI, Wageningen University, EUMETSAT and NASA-researchers. The product is thus partly independent, and partly reflects the best-practices consensus from a larger group of authors.

P7 –Lines 21-22: Maybe a copy&paste error? What does that sentence mean anyhow, I am not really sure (too vague).

This sentence should not have been there. We removed it.

P7 L30: Please make clear that the impact of transmission function is only of importance for the large spectral windows as needed for GOME-2.

We now state that accounting for spectrally varying transmission is relevant for the wide GOME-2 window.

P8-Lines6++: If I look at the list of changes made here, they really look rather incremental, fine-tuning some retrieval settings. The big question is whether this warrants publication in a peer reviewed journal as original work. You will have to justify this to some degree. (i.e. why is this more than an internal tech report?)

The list does not sum up the all the retrieval changes in this paper. Besides the narrower fitting window based on the sensitivity tests – including the exclusion of the O2-band as recommended by the reviewer – we now use a set of PCs based on a multiyear analysis of spectra over the Sahara rather than based on the last 12 months as was done before. We also implement a correction for degradation of the level-1 spectra over time, and show how this stabilizes the retrievals, which has not been done before.

P8-Line12: I think it is important to underline (again) that the exclusion of the $O_2$-A band actually helps the retrieval. This is the topic of a long-standing debate and the original algorithm included the $O_2$-A band under the assumption that it "helps" the retrieval. It would be good to elaborate more on that specific issue and show the community clearly why it harms the actual retrieval. This could be a valuable addition.

Thank you for bringing this up. This was exactly the motivation to do the sensitivity tests with DISAMAR, as described in the original manuscript (P8L9-13): "In SIFTER v1, a wide fitting window was selected (712-783 nm), which includes spectral features from the oxygen-A band (759-769 nm) and water vapour absorption (714-734 nm). These features potentially complicate the calculation of the transmittance terms with the principal component method. Here we investigate the possibility to reduce the number of PCs $f_k(\lambda)$ by selecting a narrower window that includes the strong fluorescence signature, but excludes the adjacent $O_2$-A and water vapour features."

We performed tests to highlight that exclusion of the $O_2$-A band improves the accuracy of the retrieval. In our end-to-end test, we specified fluorescence to be 4.0 mW $m^{-2}$ $nm^{-1}$ $sr^{-1}$, and then attempted to reproduce the signal from the DISAMAR TOA spectra for different retrieval scenarios and for three spectral windows, one excluding the $O_2$-A band, one excluding both the $O_2$-A band and much of the $H_2O$ absorption, and one wide window (the original SIFTER v1 window). Table 1 below shows that the accuracy (bias) and uncertainty of the retrievals drastically improves by excluding the $O_2$-A band, and improves further by limiting the influence of $H_2O$.

**Table 1.** Results of tests to reproduce a SIF signal of 4.0 mW $m^{-2}$ $nm^{-1}$ $sr^{-1}$ from an ensemble of 200 DISAMAR top-of-atmosphere spectra with different retrieval geometries, surface albedo values, and water vapour conditions.

| | Window | Bias | Uncertainty |
|---|---|---|---|
| | | (mW $m^{-2}$ $nm^{-1}$ $sr^{-1}$ ) | (mW $m^{-2}$ $nm^{-1}$ $sr^{-1}$) |
| Original window [Sanders et al., 2016] | 712-783 nm | -0.47 | 0.53 |
| Excluding $O_2$-A band | 712-758 nm | -0.23 | 0.41 |
| Including $O_2$-A band | 734-783 nm | -0.49 | 0.57 |
| Excluding $O_2$-A and $H_2O$ (this work) | 734-758 nm | 0.00 | 0.39 |

Inclusion of the $O_2$-A band tends to lead to an underestimation of retrieved fluorescence. Proportionally similar underestimates occurs for different fluorescence signal strengths.

As to why including the $O_2$-A band harms the retrieval of SIF from space, we did an additional study into the sensitivity of top-of-atmosphere radiance to SIF at the surface, and included this in the revised manuscript in Section 3.2. We simulated TOA radiances for two ensembles: one without SIF and one with a SIF strength of 4.0 mW $m^{-2}$ $nm^{-1}$ $sr^{-1}$ (at 737 nm). The DISAMAR radiative transfer model accounts for absorption by $H_2O$ and $O_2$, and describes the effects of multiple scattering and a spectrally varying surface albedo. DISAMAR radiances have been convolved with the GOME-2 slit function (width ∼0.5 nm). The settings in DISAMAR were such that the ensemble average surface albedo, surface pressure, and viewing geometry were the same, so the essential difference between the two ensembles is in the presence of a SIF signal. No clouds or aerosols were included in the simulations.

The DISAMAR simulations show that the presence of a SIF signal leads to a small addition of radiance across the spectrum. The surplus radiance closely follows the magnitude and spectral shape of the fluorescence source spectrum between 740-758 nm, but is weaker between 734-740 nm and 759-766 nm, where water vapour and oxygen partly absorb the SIF signal travelling from the Earth's surface towards the sensor (upper panel of the new Figure 3 in the revised manuscript, included here below). The

sensitivity of the radiances to changes in the 'state' thus shows a strong spectral dependence. Put simply, within the $O_2$-A band transmission is low and only half of the SIF signal makes it to the sensor. But between 740 nm and 758 nm (and also for 768-783 nm), transmission is high and almost all SIF photons reach the sensor.

The SIF AMFs calculated from the DISAMAR model between 742 and 758 nm are indeed close to 1 (Figure 1 below revised Figure 3), demonstrating the good sensitivity to fluorescence in this spectral range for our ensemble. Between 734-742 nm the AMF has values close to 0.9, and within the $O_2$-A absorption band the AMF drops to values of $\sim$0.5.

This explains why a spectral fit with a wide spectral window that includes the $O_2$-A band will not reproduce but rather underestimate the SIF signal prescribed in the simulations. The spectral fitting procedure attempts to match all spectral features within the window. For the wide window this comprises the in-filling of the Fraunhofer lines in spectral regions where sensitivity to SIF is close to 1, but also the SIF in-filling of the $O_2$-A band, where sensitivity to SIF drops to 0.5. The single, 'window mean' retrieved SIF value then becomes a trade-off between partial SIF in-filling within the $O_2$-A band and complete in-filling outside the absorption bands. The result is a compromise, a structural underestimate of SIF. The lower panel shows the diagnosis: much larger average spectral residuals (model minus observation) for the 734-783 nm window in grey, especially around the Fraunhofer features, than in the 734-758 nm window (in black).

The above explanation and Figure 3 has now been included in the manuscript (Section 3.2).

P13-Lines 25 and around: Again, you are converging to a similar fitting window as Joiner et al did for a long time already. The only thing new is that you provide some more tests (of which I'm sure Joiner did as well).

See above. We now provide more insight in Section 3.2 into the reasons why excluding the $O_2$-A band leads to a more accurate SIF retrieval. As said before, the other new

elements of our retrieval are the PCs based on a multi-year analysis of spectra over the Sahara rather than based on the last 12 months, and the correction for degradation of the GOME-2A level-1 spectra over time.

Page 15, lines 4++ Temperature can have a profound effect on water vapour absorptions (due to a wide range of lower state energies), just using the Sahara might underrepresent this change in spectroscopy.

This is exactly one of the reasons why we take the ensemble of all 2007-2012 spectra recorded over the Sahara to base our PCs on. PCs based on such a 6-year period comprise relatively high water vapour absorptions compared to a reference PC set based on the last 12 months only. Still, we acknowledge that the variability of $H_2O$ in the reference spectra is likely too low for the variability encountered in GOME-2A spectra over moist tropical forests. Our AMF analysis above suggests that this could be prevented for future improvements by narrowing down the spectral window even further to 740-758 nm, but this should be the focus of future work, as now discussed at the end of Section 3.2.

P18 L1++ Why should the bias depend on latitude at all (and not, say Air Mass Factor alone)? 1 degree also sounds really fine. What I am missing in most discussions is the lack of mechanistic motivation for certain choices. Why is the latitude dependent bias not symmetric? Why do you assume it is there in the first place?

The issue was also raised by ref1. The GOME-2A slit function is known to change significantly over time because of temperature changes [Munro et al., 2016]. Changes in the shape of the slit function are known to have caused highly structured spectral responses and thereby problems with the fitting of HCHO, $O_3$, and $NO_2$ in the UV-Vis part of the GOME-2A spectra (e.g. De Smedt et al. [2012], Miles et al. [2015], Azam and Richter [2015], Beirle et al. 2017]). These changes occur along an orbit (thus with latitude) and (we now include in the manuscript text) *"appear as an increase in the width of the slit function, with implications for the depth of the Fraunhofer structures:*

*the wider the slit, the less deep the Fraunhofer lines. Shallower Fraunhofer lines may then be interpreted by the fitting algorithm to have been caused by fluorescence, which explains the positive fluorescence bias in the southern hemisphere (Figure 4). Conversely, sharper and deeper Fraunhofer lines (relative to the width of the lines over the Sahara), may well be interpreted as caused by negative fluorescence, explaining the negative bias for latitudes north of the Sahara.“*

The problems have been mitigated in the trace gas algorithms by extending the fitting approach with dynamical fit parameters describing the width and shape of the slit function, with good results [Beirle et al., 2017]. Such an approach could also be attempted for GOME-2 SIF retrievals in the far-red part of the spectrum. These are limited by the reference spectra taken over the middle of the orbit, when the slit function has an intermediate width that is likely not representative for smaller widths north of 30°N, and the larger widths in the southern hemisphere.

P21, Figure 6: I don't understand what this figure is telling me. Why "should" the uncertainty depend on the SIF signal? What is being tested here? I am a bit lost. What would make sense is to plot the uncertainty against continuum level radiance, thus to the overall SNR of the spectrum. Absolute SIF will get noisier at higher signals (which can then lead to some correlations of SIF uncertainty and SIF signal as vegetation is very bright in the NIR). Here, however the single pixel sigma is used, which is so large that it is hard to see these effects.

Figure 6 (now Figure 7) shows that the uncertainty is not driven by the SIF signal itself, which is reassuring. This was already discussed in the text. Thanks for the suggestion to also show the uncertainty vs. reflectance level. We now do that in the new Figure 7(b).

Section 5.2: Wo what does this tell us? It is all purely descriptive. Sentences like "We find that both data products capture the seasonality of SIF, which suggests that actual fluorescence in response to photosynthesis is being measured" are rather vague. Is

the final key point of the paper that you managed to reproduce the Joiner retrievals?

Section 5.2 presents a quantitative intercomparison of SIFTER v2 against the NASA SIF product. Such intercomparisons are useful in assessing the mutual consistency of satellite retrievals, and to evaluate the SIF values and their uncertainties. A previous intercomparison of SIFTER [Sanders et al., 2016] and NASA showed large discrepancies, apparent as a suspicious 'fluorescence hole' over tropical forests in SIFTER against NASA, which was one of the drivers of the current study, which is the result of a longer collaboration between WUR, KNMI, NASA, and EUMETSAT. Section 5.2 clearly shows that this discrepancy is now gone. Moreover the comparison of the matched-up SIF pixels expressed as probability distribution of differences provides a welcome assessment of the combined statistical uncertainties from both products. That the combined uncertainties quantitatively match with their anticipated values based on the theoretical uncertainties further underlines that both SIF products and their estimated uncertainties make sense, despite the two algorithms being different in their PC-sets from different regions and time periods. We shortened the sentence that the reviewer finds vague to *"Both data products capture the seasonality of SIF."* The last word on retrieval quality is obviously with validation, which is underway for SIFTER, but beyond the scope of this study.

**References**

Azam, F., and Richter, A.: GOME-2 on MetOp-B Follow-on analysis of GOME2 in orbit degradation, Final Report EUM/CO/09/4600000696/RM, EUMETSAT, http://www.iup.uni-bremen.de/doas/reports/gome2_degradation_follow_up_final_report.pdf, last access 8 June 2020, 2015.

Beirle, S., Lampel, J., Lerot, C., Sihler, H., and Wagner, T.: Parameterizing the instrumental spectral response function and its changes by a super-Gaussian and its derivatives, Atmos. Meas. Tech., 10, 581–598, https://doi.org/10.5194/amt-10-581-

2017, 2017.

De Smedt, I., Van Roozendael, M., Stavrakou, T., Müller, J.-F., Lerot, C., Theys, N., Valks, P., Hao, N., and van der A, R.: Improved retrieval of global tropospheric formaldehyde columns from GOME-2/MetOp-A addressing noise reduction and instrumental degradation issues, Atmos. Meas. Tech., 5, 2933–2949, https://doi.org/10.5194/amt-5-2933-2012, 2012.

Eskes, H. J. and Boersma, K. F.: Averaging kernels for DOAS total-column satellite retrievals, Atmos. Chem. Phys., 3, 1285–1291, https://doi.org/10.5194/acp-3-1285-2003, 2003.

Frankenberg, C., and Berry, J.: 3.10 - Solar Induced Chlorophyll Fluorescence: Origins, Relation to Photosynthesis and Retrieval, Comprehensive Remote Sensing, Elsevier, pp 143-162, ISBN 9780128032213, https://doi.org/10.1016/B978-0-12-409548-9.10632-3, 2018.

Magney, T. S., Frankenberg, C., Köhler, P., North, G., Davis, T. S., Dold, C., et al.: Disentangling changes in the spectral shape of chlorophyll fluorescence: Implications for remote sensing of photosynthesis. Journal of Geophysical Research: Biogeosciences, 124, 1491– 1507. https://doi.org/10.1029/2019JG005029, 2019.

Miles, G. M., Siddans, R., Kerridge, B. J., Latter, B. G., and Richards, N. A. D.: Tropospheric ozone and ozone profiles retrieved from GOME-2 and their validation, Atmos. Meas. Tech., 8, 385–398, https://doi.org/10.5194/amt-8-385-2015, 2015.

[Figure]

**Fig. 1.** Upper panel: Difference between ensemble average simulations of TOA radiances with and without a SIF signal at the surface (black line). Middle panel: SIF AMF as a function of wavelength.